# A Comprehensive Review on Smart Grids: Challenges and Opportunities

**DOI:** 10.3390/s21216978

**Published:** 2021-10-21

**Authors:** Jesús Jaime Moreno Escobar, Oswaldo Morales Matamoros, Ricardo Tejeida Padilla, Ixchel Lina Reyes, Hugo Quintana Espinosa

**Affiliations:** 1Escuela Superior de Ingeniería Mecánica y Eléctrica, Unidad Zacatenco, Instituto Politécnico Nacional, Ciudad de México 07340, Mexico; omoralesm@ipn.mx (O.M.M.); ilinar1600@egresado.ipn.mx (I.L.R.); hquintana@ipn.mx (H.Q.E.); 2Escuela Superior de Turismo, Instituto Politécnico Nacional, Ciudad de México 07630, Mexico; rtejeidap@ipn.mx

**Keywords:** smart sensors, Smart Grids, advanced sensors, smart metering, Industrial Internet of Things, applications in power distribution lines

## Abstract

Recently, the operation of distribution systems does not depend on the state or utility based on centralized procedures, but rather the decentralization of the decisions of the distribution companies whose objectives are the efficiency of interconnectivity. Therefore, distribution companies are exposed to greater risks, and due to this, the need to make decisions based on increasingly reliable models has grown up considerably. Therefore, we present a survey of key aspects, technologies, protocols, and case studies of the current and future trend of Smart Grids. This work proposes a taxonomy of a large number of technologies in Smart Grids and their applications in scenarios of Smart Networks, Neural Networks, Blockchain, Industrial Internet of Things, or Software-Defined Networks. Therefore, this work summarizes the main features of 94 research articles ranging the last four years. We classify these survey, according Smart Grid Network Topologies, because it can group as the main axis the sensors applied to Smart Grids, as it shows us the interconnection forms generalization of the Smart Networks with respect to the sensors found in a home or industry.

## 1. Introduction

The transformation of the electrical energy system is taking place all over the world, moving from a conventional unidirectional structure to a more open one, configurable and participatory structure by consumers and other actors in the sector. This change arises from several motivations, differing from one country to another. From the year of 2010, the electricity industry has undergone important changes towards the use and implementation of new technologies with the aim of allowing better use and greater efficiency in the generation, transmission, and distribution of electricity. In many places, these changes have culminated in the emergence of a larger electricity market [1].

Li et al. [2] establish as motivation the satisfaction of demand with the maximum reliability and quality of service, with minimum economic and ecological costs, at the same time having maximum safety for both people and equipment used, and always keeping the voltage and frequency values within the permitted limits. However, because of the evolution of the electrical energy demand, the distribution companies make a huge effort for matching all the consumers’ requirements for a modern electrical system demand of both in quantity and quality, due to the side effects of the COVID19 pandemic [3,4].

In this new context, the operation of distribution systems, depending in some countries on local government and others on private companies, is based on centralized procedures. According to Hussain et al. [4], electrical systems tend to decentralize the decisions of distribution companies whose objectives are to maximize customer satisfaction. Therefore, government or corporate distribution companies are exposed to greater risks provoked by the need to make decisions based on hybrid distribution models that are increasingly reliable.

On the one hand, the main challenges to overcome in the electricity sector are scarcity of resources, fluctuations in the price of oil market, new forms of energy production, and the current inefficiency of the distribution systems [5]. In emerging economies like Mexico, these challenges increase dramatically. On the other hand, in terms of quality, electricity companies, mainly in distribution, face the need to comply with all the technical parameters required for reducing the number of supply interruptions. In order to improve the operation of the electrical distribution system, one of the proposed actions is to apply an integration in the generation of electrical energy through ecologically sustainable energy sources or clean energies. Thus, the use of these renewable energies will make it possible to increase energy efficiency in all subsectors of the electrical system, as long as advanced and intelligent technologies are integrated for distribution, monitoring, and management of the electrical network [6].

For an electrical distribution system evolving and adapting rapidly to variations in the demand for electrical energy, the available resources must be optimized via intelligent technologies called Smart Grids. In this sense, an Intelligent Electric Power Distribution Network or Smart Grid is a network that intelligently integrates new technologies to improve the monitoring and control of the operation of electrical systems; specifically, in generation, distribution, in addition to being able to incorporate the users’ actions connected to it. These networks are characterized by implementing, within the system, innovative equipment and services, new communication, control, monitoring, and self-diagnosis technologies [7]. Figure 1 shows the main factors that influence the composition of a Smart Grid.

According to the above, these smart energy distribution grids have developed in countries where natural energy resources, such as oil, coal, or gas, are scarce.

In the particular case of Mexico’s electricity system, it presents better characteristics than those of most of the region. However, it is far below what could be considered optimal operation, mainly in terms of electrical losses, integration rate of renewable energies, efficiency of resource management, and quality of provided services. In Mexico and in general in Central America, the main objective has been to integrate as much generation as possible based on renewable energy sources. Specifically, it is desired for the year 2030 to obtain at least 50% of the electrical energy generated from large plants by making use of renewable sources, as well as with distributed generation and micro generation by using small units for residential purposes [8,9].

To achieve the above objective, any country with a tendency towards sustainable development must have an electricity system capable of responding adequately, dynamically, and intelligently to changes in infrastructure and especially in demand. This guarantees not only energy security, but also a growing quality in the technical, economic, and environmental performance indices. Therefore, Smart Grid technologies are the ideal means to take the electric power sector into a new era of reliability, availability, and efficiency, improving the world economy and taking care of the environment. Nonetheless, during the transition period it is necessary to carry out tests, apply technological improvements, educate the consumer, develop standards and regulations, and share information among those who work in the electricity sector in order to ensure that the expected benefits of Smart Grids will be a reality [10].

Therefore, it is necessary to foresee the real impact that the various technologies associated with this change will have, both for electricity companies and for consumers and agents that interact in the electricity sector. Therefore, we have reviewed the state-of-the-art technologies in Smart Grids arising, taking into account their benefits over others and what they can provide to a functional electrical distribution system and considering all those challenges to be implemented. Thus, our main objective is to study and contribute to the knowledge and impact of these technologies in their various characteristics such as improvement of safety, reliability, quality of the electrical system, and implementation in various environments [11].

According to the above, this review is divided into six sections. In Section 2, the 94 most recent algorithms on the topic of Smart Grids are classified, finding a common taxonomy to order the nature of their work. In Section 3, entitled Current Smart Grid Algorithms, the main characteristics of each of the 94 algorithms and their contributions are exposed, and in Section 4 these characteristics are discussed. The future and trends of this kind of technologies in the intelligent distribution of energy are discussed in Section 5. Finally, Section 6 summarizes the main findings and conclusions of this survey.

## 2. Taxonomy

The operation of Smart Grids includes the use of software, hardware, and technologies that help electricity companies to identify and instantly correct imbalances between generation and demand in order to improve service quality, increase energy reliability and reduce costs [12]. There are several ways to represent the Smart Grids operating scheme. The Smart Grid Interoperability Panel (SGIP) and National Institute for Standards and Technology (NIST) designed an algorithm or methodology for organizing and/or planning the different interconnections of a Smart Grid network, describing their seven main domains: service providers, transmission, generation, customer, distribution, market, and operators. This scheme is represented in Figure 2, and shows several reasons for discussion about the characteristics, behaviors, uses, interfaces, requirements, and concepts of an intelligent network [13,14].

Different areas of current technology, in terms of Smart Grid schemes, cover the whole electrical system, from generation to different types of consumers. Several of these schemes can be considered well-known technologies with proven efficiency, but some field of this technology can be better developed because of their application for the final costumer, as many schemes still need the evolution of itself developing. A fully optimized electrical system should have technologies across the board. As shown in Figure 3, these technologies allow monitoring and control, enables information and communication technologies integration, distributed generation by means of unifying along with management simplification in terms of distribution and transmission networks. This allows not only integration of smart sensors, but also adding usage of vehicles with a certain type of primary intelligence, leading us to depict these kind of infrastructure as a management system both for all industrial and residential customers [15].

Key role Smart Grid development is played by the communication technologies, as it is necessary to manage a huge amount of information that can be coming from assorted applications that also use should be monitored and analyzed, in order to obtain a real-time response [16]. The great challenge for energy companies is to define the communication requirements and determine the best communication technology to manage the data and respond by ensuring a safe, low-cost, and reliable service for the entire system. Actually, there are several communication technologies that allow the connection among smart meters, sensors and the control center [17,18].

There are so many ways to build-up a taxonomy about the different algorithms existing nowadays. Unlike the works done by Mollah et al. [19], Ghosal and Conti [20], or Hu et al. [21], we contemplate a taxonomy of nine classifications, made-up by 59 sub-classifications overall. Each sub-classification is assigned to algorithms or schemes found in the current literature and described in Section 3.

In this section, we propose the following nine classifications as well as their 59 sub-classification and the works found in the literature:**Smart Grid Network Topology**(a)Neighborhood Area Networks (NAN)(b)Software-Defined Networks (SDN)(c)Interdependent Networks (IN)(d)Field Area Networks (FAN)(e)Wireless Sensor Networks (WSN)(f)Not Defined**Smart Grid Technology**(a)Blockchain(b)Reinforcement Learning(c)Industrial Internet of Things(d)Internet of Things(e)Machine learning(f)Data mining(g)Machine learning and neural training(h)Short-term memory network(i)Power Line Communication Technology(j)Power electronics(k)Big data(l)Fog Cloud computing(m)Energy Storage and Power Electronics Technologies**Encryption used in Smart Grids**(a)Multidimensional Data aggregation(b)Cognitive Risk Control(c)Not Defined**Type of current transmitted by the Smart Grid**(a)Direct Current(b)Alternate Current**Data Transmission over a Smart Grids**(a)Yes(b)No**Applications of Smart Grids**(a)Advanced Metering Infrastructure (AMI)(b)Substation Automation (SA)(c)Distributed Automation (DA)(d)Distributed Energy Resources (DER)(e)Teleprotection (TP)(f)Anomaly Detection (AD)(g)Privacy Preserving (PP)**Connectivity used in the Smart Grid**(a)Ethernet(b)PDH/SDH(c)WDM/DWDM(d)Fi-Wi/RoF/C-RAN(e)2G/3G/4G(f)5G(g)MPLS(h)QoS(i)WSN**Tools used for the analysis of Smart Grids**(a)Time series Analysis(b)Regression Model(c)Not Defined**Protocols Applied in Smart Grid Algorithms**(a)Green-RPL(b)Local positive degree coupling(c)IEEE 802.11s(d)Web Of Energy(e)Dynamic Barrier Coverage(f)IEC61850(g)Wind-driven bacterian foraging algorithm(h)Data Slicing(i)TSUBE energy trading algorithm(j)Stochastic Geometry(k)Rectangular quadrature amplitude modulation(l)Policy-based group authentication algorithm(m)Mapping interface integration COIIoT(n)Nash Equilibrium (NE) and the Bayesian NE(o)Wireless sensor network protocol(p)Algorithmic Approach

We consider that the first classification, Smart Grid Network Topology, can group as the main axis the sensors applied to Smart Networks, as it shows us that the interconnection forms generalization of the Smart Networks with respect to the sensors found in a home or industry.

## 3. Current Smart Grids Algorithms

### 3.1. Neighborhood Area Networks

Aladdin et al. [22], develop a multi-agent reinforcement learning model (MARLA-SG) scheme for improving the efficiency response into energy demand of mart grids (SGs), being an alternative to the enormous growth of urban settlements, which have become large in extension and very dense in population. Traditional connection topologies are today not very adaptable and are no longer compatible with the dynamism of current electricity consumption. Nowadays, an intelligent network is necessary not only to modify its topology, but is also to be based on intelligent demand management so that it adapts to changes at all times. Peak-to-Average Ratio (PAR) with its respective cost reduction is the main goal of every designer of this type of smart energy grid. This scheme bases its worth on the simplicity and flexibility of choosing or discarding elements along the network. Its operation is based mainly on the Q-learning and State-Action-Reward-State-Action (SARSA) schemes for reducing PAR by 12.16% and 9.6%, while the average cost fell 7.8% and 10.2%, respectively.

Almshari et al. [23], propose network security as an important point nowadays because of the many requirements to be match by the customer such as the privacy of the most user’s sensitive data of an electrical network. The whole of sensors or smart devices that make up an intelligent network consume energy, as well as share their information among the members of the network, is considered as an IoT network. Thus, IoT networks must inform higher-layer applications of their energy consumption as well as the data and information that is shared whether it is sensitive or not. Therefore, the energy consumed by each execution or request of a given application would be managed more efficiently. This administration would be successful as long as individual consumption profiles are generated by device and the tasks that are carried out for detecting normal consumption, but they can be infected by malicious software such as a computer virus or a generalized system failure of a computer. The experiment proposed in this work is to continuously measure the average consumption of two computers, both with the same tasks but one infected with a virus, what this experiment yielded is that the energy is affected when a computer executes unnecessary tasks or because of some tasks that make the operation of the system unstable.

For the operation of intelligence networks is essential to manage a large amount of data over the network, enlightening the importance of NANs (Neighborhood Area Networks), whose main objective is to intelligently control the traffic generated by consumers towards the control command. The flexibility and dynamism of wireless networks is a point to consider when implementing this technology in NANs. Thus, the work developed by Astudillo León and De la Cruz Llopis [24] modifies in a general way the routing protocols of these wireless networks, using a multi-hop mesh. The authors modify the protocols to add the possibility that the information takes different paths or possible channels of information distribution depending on the needs of the entire network. The NS-3 simulator is used to evaluate the new load conditions and therefore the generated traffic. This proposal shows great results in terms of performance and transit time of the information on the proposed network.

The use of IoT protocols and algorithms in SGs has had a growing boom and solved problems due to the intelligent use of data and information generated in current networks, even when they are dedicated to the distribution and administration of electrical energy, as current demands require offering the end user associated services that increase needs, reliability, and efficiency. Cluster analytics has the ability to process a huge amount of data generated in these SGs, making use of artificial intelligence tools such as supervised learning. The disadvantage of this type of intelligent computational tools is that it sometimes exposes the private information of the users. Therefore, Guan et al. [25], analyze the information of these clusters, preserving the privacy of the users via the IDPC algorithm, which is a private cluster based on an IGMM model, i.e., a Gaussian Mixture Infinite. The IDPC proposal combines a differential privacy model with the nonparametric Bayesian method. The latter allows certain parameters to change dynamically with the data and in general this algorithm groups an infinite number of possible groupings. To make IDPC differentially private, the authors propose to use a Laplace mechanism in order to release the data.

Haghighat et al. [26], present a market model to integrate various programs for managing energy in SGs that operate independently. This model uses a two-level optimization system, i.e., it operates in low and high layers of a general market model. Thus, the authors propose that entities in low-level SGs are independent and autonomous from decision-making in a higher layer market model, which is usually unpredictable given the random flow of alternating current over the network. Thus, a system is generated with a single leader who is followed by multiple followers in order to obtain the best moments throughout the day to optimize the renewable resources within the network, the response demands, or the various types of storage devices, all of which are accomplished via linear programming, known as mixed integers. This type of programming is used by the two layers of the dialing model to be optimized and communicated.

Joseph and Balachandra [27], introduce the Energy Internet concept as a new way of exchanging affirmation about SGs, shifting the paradigm in the interconnection between users who consume energy and those who provide, generate, store, and distribute the electric energy. Within Energy Internet, not only does electrical energy move or distribute, but it also serves to establish communications and, above all, bidirectional communications with paradigms, algorithms, and real-time protocols. According to the authors, Energy Internet should emerge as a response and evolution to renewable energy transactions, which should use only modern transmission media but also storage and distribution. Furthermore, Energy Internet is a combination of various well-known technologies, mature in their evolution and widely used in SGs, including Blockchains, Artificial Intelligence algorithms, and tools and Internet of Things protocols. Therefore, the main approach of that work is to study the feasibility of Energy Internet in the electricity generation and distribution system. Initially, the authors make a summary of the SGs and their used tools in order to take the best tools, protocols, and algorithms from the state-of-the-art, and thus to explain the successful inclusion of the Energy Internet in the growing energy market. Last, the authors conclude that current energy systems are ready to use Energy Internet in terms of infrastructure but lack public policies, which would greatly limit their implementation.

By the year 2021, in the world only 5% of the vehicles that circulate on the streets are electric but their development and use in end users has been growing constantly in this decade, since awareness has been raised by the consumers of the atmospheric and health implications of the use of vehicles that run on non-renewable fossil fuels. This increase has not been even in all the nations of the world, in the fastest growing countries the need arose for charging stations that sell quality energy. These service stations have problems usually due to harmonic distortions caused when the electrical energy is distributed by the electrical networks. Thus, the main proposal of that article proposed by Khan et al. [28] lies in an efficient model of charging the batteries used in electric vehicles from sources of random or unpredictable generation such as alternative solar and wind energy. This model aims to control the duty cycle in an efficient way for stabilizing and regulating the constant voltage management, i.e., for making the conversion process from high voltage DC to low voltage DC in an efficient way.

Another work that faces the inclusion of full-duplex or bidirectional interactions is the one made by Montanari et al. [29]. These interactions optimize and greatly help the operation and especially the maintenance from the point of view of how the network is managed in terms of to their assets. These maintenance and management actions use tools for diagnoses, which are costly both economically and computationally speaking, as intelligent monitoring is a very important task but can rarely be performed in real time. Therefore, this article states that information must be acquired starting from the intrinsic components of the electrical network itself. Partial downloads can be considered one of the most harmful elements that the network can handle, so the authors propose a way to recognize these downloads using artificial intelligence tools. Thus, the evaluation carried out by the smart sensors along the smart grid will help to estimate the partial discharges and the continuous verification of the components found along the electrical grid. The sensors can carry out this evaluation both automatically and dynamically trying to maintain the most optimal conditions of the network over time, which increases reliability and makes the initial investment return optimally.

According to Wang et al. [30], the evolution from conventional cities to smart cities lies in large implementations of Internet of Things (IoT) protocols, reducing greatly the dissipation of the energy consumed. The use of the IoT in conventional energy networks makes them SGs containing sensitive information, as multiple consumer bills are sent and received from users. Thereafter, malicious users can train entities based on Q-learning algorithms that know the consumption profiles before the service providers. Thus, the main proposal of the article is to define a simple Q-learning algorithm in its operation but that preserves the privacy, which they call LiPSG for managing energy in these SGs. The LiPSG scheme first separates into uniform parts whose order is random, then to prevent sensitive data from being transmitted and exposed and thereby compromising the privacy of users. The authors are only using in LiPSG a function that exchanges hidden messages but also the technology called Edge Computing, which helps in an important way to increase the efficiency of the algorithm.

Another algorithm that uses IoT protocols is the one developed by Xu et al. [31] without doing a redesign that modernizes the conventional electricity grid. This modernization occurs in terms of efficiency, security, and reliability, as it improves the distribution and transmission systems to the end users. The problem addressed by the article lies in the urgent challenges any smart grid has, e.g., the minimization of transmission costs and the renewable resources distributed throughout the network along with their respective storage media. Therefore, the authors analyze and model all the characteristics and factors influencing the SG in a distributed manner such as the service reliability and the cumulative cost of electrical energy, as well as their impact on certain means of storage and infrastructure integration. Information networks play a very important role since it is possible to measure the traffic supported by the network by means of the demand estimation.

Smart grids are vulnerable to various types of cyber-attacks due to their high requirements of communication networks to convey, sense and control data to improve its efficiency in matters as: energy generation, transmission, and delivery. Alnasser and Sun [32] suggested a fuzzy logic trust model to locate malicious nodes in SGs networks, contemplating unstable behaviors that impact the model such as contradictory behavior attack and on–off attack. Their simulations results pointed out an upgrade of the packet dropping rate by up to 90% in the cases where there were 25% or less of malicious nodes, comparing it with the Lightweight and Dependable Trust System model. They conclude that their model gives network designers a complete bundle that offers dependable messages with high reliability via a secure route, and it could be employed to Vehicular ad hoc Network with mobility factor.

Route instability is one of the causes of packet loss which affects network reliability in a SG. Therefore, Hsieh and Lai [33], have used of IEE802.11s as a solution to deliver high speed and reliable data transmission in neighborhood area networks (NANs), which are one of the three categories of SG communication networks. This category provides electric power station surveillance, condition monitoring, and management as they function as a backbone between home area networks (HANs) and substations of an electrical company supplier. Therefore, they must supply a high-quality service (QoS) for time-critical data and identify between several types of data to deliver the large numbers of data packets to central servers at high speed and with a reliable transmission. To match these requirements, IEE 802.11s uses an enhanced distribution channel access (EDCA) to provide QoS, as well has a multi-hop routing protocol Hybrid Wireless Mesh Protocol (HWMP) and unique topology formation.

The correct communication of data in neighborhood area networks is critical for performance and secure stability on SGs and avoid any problem in distributed energy resources to handle fluctuations demands of energy from users. To achieve this, Asuhaimi et al. [34] propose integrate heterogeneous cellular networks (HetNets) for simultaneous transmission of SG neighborhood area networks data which use a power control scheme and distributed intelligent channel access to get maximal energy efficiency and comply with delay constraints, as well as phasor measurement units (PMUs) trained with deep reinforcement learning method (DRL) with minimum interference to macrocell and small cell users and signal to interference plus noise ratio requirements. The PMUs can learn systems dynamics and establish optimal policy with excellent performance in any given number of users even if the PMU does not know about systems dynamic in advance.

In order to reduce the costs associated with electrical energy and avoid maximum demands, Apaydin-Özkan [35] present the AS-REMS model, based on the main programming of home appliances connected in the internal SG. AS-REMS is a system designed to manage mainly residential consumption, and it is established from the starting effects of the connected electrical appliances and the variations when they are being used. Devices and sensors connected to the network can be of two types: (i) Centrally Controlled (CC), which are programmed by centralized nodes and processes, and (ii) Controlled by User (CU), which accept manual configurations by a user. Therefore, the management of the load model in the SG is monitored and recorded, including habits and preferences of use of the sensors and connected devices every day. To model the system, CC devices are monitored to estimate the preferred hours of use, while for CU devices they are estimated manually, and their estimation is an approximation.

The IEEE 802.15.4 standard is responsible for establishing a centralized communication infrastructure within IoT networks. The same standard defines LR-WPAN, an intelligent wireless personal area network that works at low speed. This standard uses CSMA/CA for carrier sense multiple access, which prevents collisions. Sensors and devices counted to any smart network are prone to attacks on the internet. In smart IoT networks, DoS attacks are the most aggressive, as they prevent legitimate communication between sensors. Thus, Sadek et al. [36] propose a model that is based on a mechanism to estimate a threshold that effectively identifies greedy and malicious nodes with 99.5%.

Padhan et al. [37] analyzed average symbol error rate (ASER) and average channel capacity (ACC) performance for dynamic home area network (HAN) SG communication system, due to their significant relevance to manage the transmission of data through a smart meter between communicating devices. This article derived mathematical expressions for ASER using rectangular quadrature amplitude modulation (RQAM) and Gaussian minimum shift keying (GMSK) modulation over Saleh–Valenzuela (S-V) and Weibull fading channels to be suitable models for communication channels that are indoors. Many results proved the effect of traffic intensity, the quantity of active devices in HAN, and the modulation types with parameters of practical interest. Furthermore, they found that in ACC the fading channels relies on the intensity of the traffic, number of users, and the declining parameters; in a RQAM modulation scheme there is a better execution when the quadrature to in-phase decision distance ratio is the unity; and in GMSK modulation scheme there is a larger value for both alpha and beta.

### 3.2. Software Defined Networks

The topology of the networks has been defined via physically connecting the network cables, i.e., in a ring, in serial, or using a token, but today it is possible to establish Software-Defined Networks (SDN). The introduction of SDN to the energy sector has represented a paradigm shift in the distribution of energy to the consumer because of the flexibility and dynamism of topologies. Therefore, SDN adapt to many present or future situations and do not need to change the infrastructure to adapt to new connection models. Thus, Al-Rubaye et al. [38] propose a SDN as a platform for the Industrial Internet of Things (IIoT) which adapts in real time to possible changes in the topology or energy demand. The use of the SDN paradigm is based on real-time monitoring, which can be adapted to both the needs of operators and consumers, so that not only demand but also consumption can be managed. This algorithm works using a SDN switch which manages the resources in a local network, which adds IIoT nodes according to the needs of the system. All of this works at the infrastructure layer, i.e., direct hub-to-hub flow of switches. Experimentation for both these topologies and IIoT-based solutions shows improvements in the residency of the network, providing it with a primary level of intelligence.

All systems tend to their evolution otherwise they will tend to their disappearance, so the electricity sector should not be the exception, that is why today this sector has made use of the IoT through Wireless Sensor Networks (WSN). Like any emerging system, its main problem is the information security exchanged throughout the entire network. Therefore, Alladi et al. [39] propose to integrate blockchain technology into an intelligent network, where the advantages and disadvantages of using blockchain to increase the security of the entire network are discussed. In this work, the blockchain and structure are mainly discussed because they are an essential part of a blockchain algorithm. These blocks have been developed to offer secure administration in the exchange of data between the peers, i.e., the consumer and the provider. Blockchain technology not only helps in security, but can also help network monitoring, thus failures can be found and predicted from this system.

Electricity demand evolves day by day leading to reliable and dynamic networks that adapt to the demands of new technological developments. This is how the new goals in this sector are to introduce sensors to the energy distribution networks so that they control it in an adaptive way. The introduction of these sensors to the network requires a stable communication capable to be fail-safe throughout the network. SGs are the technological alternative that meets current needs in this area. The work done by De Almeida et al. [40] summarizes both the technologies of academic proposals and practically applied in the energy sector. This work gives an overview of SG applications, how they are protected remotely, and certain faults are solved remotely. An outline of important points is then given in terms of the transmission and distribution of energy until reaching the end customer, covering not only the general definition, but also the connection protocols that include packet circuits and especially switching.

Given the emerging need for smart consumption at home, the need to make billing and energy charging more flexible has also been born. This adaptability will vary from user to user, but the network must adapt intelligently to the challenges posed by individual users of the network. Under this scoop, Farao et al. [41] define G2GO, i.e., a Grid-to-Go scheme which broadens the range of smart grid algorithms in the field. However, the definition of new algorithms entails new challenges, especially regarding the privacy and integrity of the data sent and received on the network. As a natural evolution of these systems, the P4G2Go algorithm is also proposed, which includes the privacy necessary to preserve the information integrity of consumers and service providers. This proposal underlies the use of Idemix-type cryptographic algorithms in order to focus their operation on not linking the data of the end consumer using anonymous credentials. To increase the robustness of this algorithm, protocols such as MASKER and FIDO2 are integrated because they are stenographic schemes that do not require complex authentication and do not use a password.

The term Distributed Energy Resources (DER) brings together a large number of other terms referring to a sustainable economy with the use of smart technologies. For the optimization of this type of system and their integration into the distribution of energy, the term of exchange of energy transactions between consumers and energy generators is used. For that optimization to be feasible, privacy between peers must first be maintained. Therefore, Gaybullaev et al. [42] adopt a blockchain technology for this exchange of energy transactions. As technology advances, and due to opting for a distributed technology dependent on network transactions, security concerns are present among users, as they do not want to expose their most sensitive data to third parties. Therefore, the agenda of these transactions is safeguarded by a DSO (Distribution System Operator), generating certain nodes in the blockchain. Next, algorithms that use Ethereum schemes are proposed, via a chain of blocks that encrypts the values, in a functional and secure way, matching the consumption pairs. The algorithm of this proposal underlies a protocol encoding non-floating numbers in a series of vectors that make a comparison of all the encrypted texts in these series. This algorithm has certain shortcomings because the processing time depends directly on the number of vectors handled, increasing drastically if the number of encrypted vectors grows and making encryption impractical at the moment. Thus, this proposal faces the challenge of time by means of a DBC or Dual Binary Coding, reducing significantly the amount of calculation carried out in the encryption phase.

Ghosal and Conti [20] point out that the next generation systems must be taken into account today due to they gradually try to change to current energy transmission technologies via the so-called AMI (Advanced Measurement Infrastructure), i.e., smart sensors applied to dynamic networks. However, AMIs are not limited only to measure, collect, and analyze all the information that circulates on the network, using artificial intelligence algorithms to fulfill this purpose. Therefore, it can be affirmed that SGs function properly in an AMI but are in some way vulnerable to attacks, so these intruders and their malicious actions have chances to generate damage to the electrical system and the opportunity to launch thousands fruitful attacks in many cases. Due to the importance of AMIs in SGs and their vulnerability, the need arises to provide them with a certain degree of security to lessen the damage caused by these attacks. An alternative is the Key Management System (KMS), used in that work due to its effectiveness in terms of security. Thus, the applications and challenges of KMS applied to AMIs generate a wide expectation of research.

Ogbodo et al. [43] introduce a more advanced term of artificial intelligence, as in their proposal, they use the CRSN that are basically a Network of Cognitive Sensors based on Radio. This is achieved through the intelligent use of the sensors of the ISM band that refers to Industrial, Scientific, and Medical, the intelligence in this algorithm lies in avoiding interference between cognitive meters. Furthermore, the SG is controlled and monitored by these sensors from the generation of electrical energy to its consumption. From the latter arises the need to define an architecture or topology based on CRSNs that is reliable and that meets all the needs requested by the modern SGs. Therefore, the authors investigate and define how to implement both the algorithms and the applications of the CRSN in an intelligent network to be a support of QoS protocols for improving the way energy is consumed by the different stages along of the network.

### 3.3. Interdependent Networks

Coupling two networks to simulate a real network is essential for an interdependent SG. Wang et al. [44] proposed a local positive degree coupling (LPDC) strategy for a coupled network, analyzing the characteristics of the power network, by considering the significant modulation property and focusing on the coupling strategy, and leveraging a community detection algorithm to make a local network. In other words, the LPDC strategy presents two related algorithms: the Assemble Community Algorithm and the Positive Degree Coupling Algorithm. The results of the experiments show that LPDC had exceeded the Complete Random Coupling Method. They claim that their strategy could be applied to a multilayer network for other researchers.

There are various computer systems based on a cloud paradigm for their services and SGs are not the exception, but they have two major challenges: (i) low latency and (ii) immediate services considered as real time, leading to the more frequent use of edge computing algorithms. Thus, the work proposed by Wang et al. [45] focuses the central problem on the inadmissibility of conditional anonymity and flexibility in key management in most cryptographic protocols, which are used to establish security. Therefore, the authors of this proposal present a blockchain-based edge computing scheme that first includes mutual authentication, then proposing a key agreement protocol, both tools are included in SGs. The blockchain takes advantage of everything related not only to block chains but also conditional anonymity schemes, thereby obtaining a more secure and efficient key management, and then avoiding the use of complex cryptographic primitives.

Ghorbanian et al. [46] present some points to enhance the SGs performance, focusing on energy trading issues, flexibility of power systems and negative pricing issues, energy management, arbitrage and pricing issues, SG management issues, renewable energy resources integration issues, and SG financial transactions, giving rise to mechanism next to financial advantages from blockchain-based cryptocurrencies utilization that could get to a flatter load profile for avoiding negative pricing and price peaks. Furthermore, Ghorbanian et al. [46] expose the need to adapt blockchain algorithms to ensure the use of cryptocurrencies by communities of Internet users eager to have more secure algorithms when carrying out transactions in the field of energy systems and then in their markets. Moreover, the authors define a scheme to show the efficient use of these digital currencies in energy systems that can be considered with a current perspective and, but especially towards the future, with the aim of anticipating any challenge to face. Thereby, the authors focus their article on two main axes: (i) current challenges and (ii) existing problems and their possible solutions. Both axes not only focus on the safety of users in the energy market when trying to carry out transactions with cryptocurrencies based on the blockchain, but also on certain management and integration criteria with high competitiveness and efficiency.

Measurement by means of advanced sensors has grown in terms of the diffusion of the infrastructure. Now the analysis about how users consume is important for companies that commercialize electrical energy and data science tools such as data mining to analyze consumption patterns. Therefore, Si et al. [47] have proposed techniques that group consumption loads to estimate how the data distribution is and thus estimate all the components in the network load as a whole, intending to develop certain techniques to be used in SGs favoring all actors in the energy sector, but especially end consumers. Thus, that article makes a compendium of the main concepts of load grouping in SGs. These concepts are summarized in a five-level taxonomy in terms of charge types along with eight of the most important estimators of validity based on the type of grouping employed in the nature of the electric charge.

In [48], the authors establish an optimization model for Time of Use or ToU programming for Residential Energy Management (REM). Thus, this model can determine the energy consumption preferences of certain kind of sensors; this is of extreme importance to manage a small internal smart grid for domestic consumption. On the one hand, the authors propose a REM model, robust enough to establish communication and exchange of vital information on the network to efficiently control the loads generated in a smart home. This energy management is based on algorithms found in the cloud layer and can be considered as microgrids with almost immediate response or in real time. On the other hand, the model estimates the costs of the ToU of the tariffs for the hours of maximum consumption, which is based on a Gray Wolf Optimization or GWO together with an ABC or Artificial Bee Colony. The Raspberry Pi3 sensor array used by these authors is based on the open-source programming of various cloud computing platforms such as Node-Red or IoE, for instance.

Peng et al. [49] developed a heterogeneous Cyber-Physical System (CPS) to study the security of the SGs due to their multiple interdependent sensor networks. They used swapping link strategies to evaluate the impact of random attacks on the security of interdependent sensor networks and cascading failures between interdependent sensor networks. Based on their experimental results, they conclude that the scale-free network is more sensitive to the security of the system by exchanging internal links than in the Erdös–Rényi network. However, the CPS needs to be more complex and interdependent empirical model of sensor networks to simulate actual network systems as well as to improve the artificial intelligence optimization algorithms.

The use of P2P protocols makes the electricity market decentralized, promising to optimize the network and reduce costs to the end customer because trading is conducted directly between generators and energy consumers. Due to the decrease in the losses or leaks of energy always present, it is necessary to stop biasing costs by reducing the elements that interact with the network administration. Therefore, it must be adapted the energy traded between peers. Thus, Paudel et al. [50] proposes a market scheme without having a central entity for energy negotiation, i.e., the way the privacy of all elements of the network is traded and protected. The grid consumption prices are estimated from the interactions and energy used by the interested peers.

An incentive-based load shedding management structure within a microgrid environment and equipped with IoT infrastructure is generated by Zaidi et al. [51], working with the principles of reverse combinatorial auction and considering multiple consumers willing to decrease their load in the peak hours to later get incentives. The properties of combinatorial auctions are used so the participants can bid in packages to maximize their overall system’s social welfare in a microgrid environment. Particle swarm optimization algorithm and hybrid genetic algorithm are applied on the proposed combinatorial auction for the winner determination problem. Simulating the stability test and the performance evaluation of the scheme via MATLAB, the authors claim that combinatorial auctions are very suitable for a maximum of 50 participants for loading shedding management. For future work, the authors suggest adding non-financial incentives with the existing financial incentives.

In the large-modernized cities the energy demand has increased significantly, causing the emergence of microgrids (small power grids with a local source of supply), which uses sensing technologies and Fog-Cloud computing infrastructures to build smart electrical grids. Barros et al. [52] used Fog-computing with low latency to reduce costumers’ bills, by controlling and managing power demand and production, thus disappearing the big bulk of data being passed to the Cloud and reducing the requests’ response time. They used real-time data from their GridLab-d simulator to know the current energy rate, using this data they made the scheduling, performing the appliance shifting it to a time of the day when the energy was cheaper. As a result, algorithm execution time can be shorted by the infrastructure, reducing the time of each request by up to 21.49% respect to the cloud, also offering low latency and real-time computing. Finally, the authors proved that it could make the calculations required to control a Microgrid, providing quicker returns and keeping a CPU utilization rate less than 50% during requests.

Yang et al. [53] developed a routing protocol for low power and lossy networks to fulfill the requirements of green communications in SGs, called Green-RPL, to lower the energy consumption in Cognitive Radio enabled Advanced Metering Infrastructure (CR-AMI) networks, considering the Energy Efficiency Over Virtual Distance (EEVD). During one hop, multiple neighbor nodes are chosen to arrange a forwarder set, and the EEVD of each one is approximated and adopted as the basis of forwarding priority. As a result, the node with the greater energy efficiency is more probable to be forwarded, thus the energy-efficient route can be picked. Furthermore, their Green-RPL guarantees security for primary users, even though satisfying the utility needs of secondary users, fulfilling the QoS requirements of communications in SGs.

### 3.4. Field Area Networks

Ahmed et al. [54] presented an Energy Management Model (EMM), which integrates both Gaussian Process Regression (GPR) and Machine Learning (ML). It consists of different stages: in the first one they integrate different models such as PES (Prosumer Energy Surplus), GR (Grid Revenue), and PEC (Prosumer Energy Cost) for training a ML model in order to obtain different base-performance parameters. In the second one, given the stochastic properties of various sources such as the load or the price of energy to the final consumer, they calculate a GPR model using these sources as variables to the system together with a genetic algorithm. As all systems have a degree of uncertainty, in the third one the authors add certain temporal parameters such as GR, PEC, and PES, with the aim of removing temporary distractors that may influence the generation and distribution of energy.

Today’s SGs encompass not only alternating current networks, but also Direct Current (DC) smart grids, due to these SGs can distribute low or high voltage with their alternating current (AC) pairs, but unlike for AC networks, DC networks can be less expensive when transmitting data or control protocols over DC power. Thus, the work made by Barnes et al. [55] mainly proposes a generalized description but focused on the data transmission on DC in high power or better known as HVDC via the ultra high voltage systems that currently prevail in commerce sources with intelligence, providing a reliable reference of the work done in DC Smart Grids.

There are other articles that include topics affecting any human society. For instance, the work proposed by Chui et al. [56] mentions the concern about the warming of our planet, generating very important changes in almost all parts of the world. Due to the above, that work is focused on reducing the energy consumption in homes by making it more intelligent, i.e., consumption is adapted and there are no unnecessary energy costs. In that work, the ELD (Electrical Line Disaggregation) is used and the electrical energy is discomposed into household appliances within a conventional home. Thus, nowadays ELDs make use of artificial intelligence and base their operation on computer science algorithms. Another important term used is the OCEEMD-WPT for optimized full set empirical model decomposition and wave packet transformation, which is introduced to model changes in power line noise to the end user. This leads to great benefits in collecting the data necessary for the network operation.

According to Fan et al. [57], a system considered modern must have various algorithms that intelligently collect data, one of them is called fine-grained in order to improve energy consumption. These authors establish that the users’ privacy depends on a trusted authority, meaning a great challenge in the exchanged information security. The blockchain is used to deal with these security problems, and then it can be considered a viable solution as long as it can be adapted to the needs of SGs. Thus, authors propose a DPPDA scheme, as they make use of decentralization via the data aggregation for preserving privacy. In addition, that algorithm uses an intelligent sensor for residential as a centralized point or node to build a block within a more complex system. Therefore, these individual blocks take a Paillier encryption system, embedding mainly consumption information on the data transmitted over the network. SHA-256 function and Boneh–Lynn–Shacham digital signature are used only for adding integrity to the data, in addition to making it more reliable when transmitted and/or shared over the network, in order to entail an optimization in the regulation of the energy consumed. This algorithm does not depend on a TTP or a CA since the scheme generates information protection by deconcentrating the data.

Hittini et al. [58] mainly design a protocol defined as FDIPP, i.e., it is a False Data Injection Prevention Protocol, which is designed to be used on a hierarchical architecture of any power distribution system that includes multiple entities. FDIPP is designed to guarantee the integrity of various parts of the system for preserving the integrity above all of the data shared by the network, which is achieved by avoiding both the access of nodes not contemplated in the network as well as the alteration or even duplication of packages. The main benefit of using this protocol is that service outages are avoided in order to reduce or eliminate damage to both the electrical network and the devices connected by the end users of the network, an this is achieved with the correct prediction of conclusions about the load on the network. Thus, the authors present a connection and communication model of a scalable Wi-Fi-based intelligent wireless network, which is endowed with a certain level of intelligence since it optimally distributes loads. This connection model then communicates certain sections of low voltage primary stations with secondary stations considered medium and high voltage.

SGs advance every day, and the conception of the technology used in these one must not only take into account algorithms and mathematical optimization models that provide intelligence to the network, but also ecological aspects must be considered such as wireless networks with reduced physical infrastructure for their benefit not only to the planet but also better to their economic returns. Thus, Hu et al. [21] present the review of a large number of algorithms and models that contemplate both planning and commercialization as well as redistribution and the ways of using energy intelligently when it circulates through these SGs. The evolution of this work begins with works of well-known models on renewable energy in terms of its collection, then a summary of intelligent technologies without a totally physical infrastructure but using optimization models is made. Finally, that work gathers articles that consider modern technologies applied to the distribution of data over the network such as 5G or B5G.

Khalifa et al. [59] define the current need to improve the traditional capacities of electricity transmission networks. To consider an electrical network as fully intelligent and automated, it must have both a high-performance basic infrastructure and a reliable and secure communication. To fulfill the latter, it is necessary to add more intelligent elements to the network both in the generation and in its administration, including all the sub-processes generated in low and medium voltage substations. Then, the information between substations will circulate more efficiently and will expand the number of purposes of the electrical network use, in terms of its automation, as long as a low propagation delay between subprocesses is kept. In addition, the authors propose the union of two emerging technologies formed by elements of different class or nature, as they merge 3G/4G wireless technologies of a mobile or cellular spirit with Wi-Fi Technology of the essence of conventional local internet networks. This is intended for communication with greater speed and reliability between the distribution points in its primary sector and especially in the secondary secondary one. This fusion of wireless technologies make this proposal of a single network that acts as if both technologies were of the same nature and no differential is perceived.

The article developed by Lee et al. [60] considers the use of sensors or smart meters to transform a power grid into a SG. These smart sensors store a large amount of information that circulates via the network, and based on this information it is possible to estimate and know what is happening in it. However, the large amount of data makes it unfeasible for the moment, given the communication and storage of thousands of IoT nodes. Therefore, an intelligent algorithm is needed to encode and order this massive amount of information, resulting in less data being needed to represent the originally compressed data. Thus, techniques are used to encode the entropy of the data via AE (Automatic Encoders), based on artificial intelligence tools, specifically deep learning. So far the properties in the data spectrum are a limitation to obtain important compression rates. Thus, the proposed compression is based on AE models, choosing frequencies that improve the recovery of the original data. The compression is optimized through spectral windows and the frequencies are chosen depending on the amount of entropy that contributes to the final result.

The work carried out by Liu et al. [61] establishes the need to observe the rapid growth of energy that does not depend on fossil fuels and lays the foundations for SGs to be equipped with the most important processing and, above all, communication capabilities. Thus, it is necessary to make a more efficient wireless network for a better management of energies considered clean or renewable. The main problem with this scheme is that wireless networks start from a star or centralization topology, i.e., a large number of secondary nodes to a single main node, increasing the amount of energy to achieve stable links and compromising the security of all secondary nodes by sharing information via the same centralized path. A possible solution to this dilemma is the use of a chain of blocks. Therefore, the authors propose an energy trading algorithm using the blockchain for smart wireless networks. With these blockchains generated with redundancy by the blockchain, it is possible to have an intelligent control and administration of the consumption of each user, as now decisions can be made based on the most convenient states of the network. Thus, the improvement in the commercialization of electrical energy depends on the efficiency in the security provided by the double chain applied by blockchain.

Tang et al. [62] address in their article the growth of social media to be used to make the management of the end user of electrical energy more efficient. However, attacks on networks so exposed to public view are constant, at the software or hardware level as well as about the information uploaded and published on social media to be prone to be malicious, imprecise information, no confirmed and that it ends up being a fake news, causing serious problems to the electrical network. Therefore, this proposal models an intelligent network system that connects to a social media by finding the price volatility from any social media. The model is based on finding the consumption information of the end user to try to reprogram it in order to know the demand and then make the distribution of the energy consumed more efficient. The programming used to meet the objective is multi-level influencing that takes into account the personal data and consumption profiles of users to prevent fake news from reaching them. Various models are also considered to respond to the different attack tactics and how an operator or distributor of the electricity grid responds.

Nowadays, the electrical networks have become not only in managing consumption energy but the things that can circulate on it have also grown such as users’ information or data or the energy provider rates, making IoT schemes emerge as an alternative to better and optimize these computational models. Thus, Seneviratne et al. [63] first evaluate the problems caused by the corrupted information circulating along of the network due to failures in transmission or quantification and even in the simple measurement of consumption. Moreover, it is stated that these failures are commonly corrected with a data retransmission, causing the networks to increase their noise and propagation delay. Thus, the proposal of that article lies in the fusion of the statistical data of smart meters, including data from consumers and suppliers without a defined structure. Therefore, the authors not only evaluate the latency and efficiency, but also the complexity and precision of this statistical fusion.

### 3.5. Wireless Sensor Networks

One issue on SGs is to get power data protection to users, as smart meters can have delays and inefficiency because of the large amount of information acquired could be stolen, compromising the safe operation of the grid. So Ming et al. [64] propose a multidimensional data aggregation scheme with an E1Gamal encryption (homomorphic) to allow gathering overall power data consumption and specific power consumption of each device of users. The above allows to achieve efficiency in communication and computation cost, making use of elliptic curve cryptography. Furthermore, this approach demonstrates via running security analysis that all the security elements required (integrity of data, privacy protection, data electricity usage confidentiality, authentication of every electrical equipment, and resistance to different kinds of attacks) were achieved. Basically, the scheme uses a system model consisting of third trust party (TTP), control center (CC), gateway (GW), and smart meter (SM).

Akerele et al. [65] propose a heterogeneous communication architecture solution that combines two or more communications systems to decrease the delay and reliability requirements of the SG for high priority traffic. From a Cross-Layer WSN-Modified Optical Coding (XWMOC) QoS mechanism of two stages and several simulations, the authors purpose Fiber-Wireless Sensor Networks (Fi-WSNs) to be used in cooperation with the Optical Network Unit (ONU), in order to decrease the end-to-end delay for the delay critical SG monitoring and control applications. On the Wireless Sensor Network (WSN), the delays of packets and in the Ethernet Passive Optical Network (EPON) could be reduced by using out of band pulses adaptive service differentiation in the queue at the ONU.

Integration of IoT on SG makes necessary and optimal bandwidth to transmit all SG data. Sandoval [66] proposes to use IoT integration for a 2400 MHz communication bandwidth with an unlicensed ISM wide band that it is found on wireless sensor networks (WSN) and Mobile Ad hoc Networks (MANET) which are part of IoT. When running TOSSIM simulations (dual band propagation model-based LNSPL model) in terms of power consumption, packet reception rate, and average network delay 2400 MHz bandwidth, it has been show a better option for small networks when there is not other devices interfering with signals. However, in the specific case of SG with IoT inclusions, where the size of the network is larger than 915 MHz band, the performance is better, no matter of mean inter-node distance and packet length. Nevertheless, the authors pointed out to considering the environment and layout of the network.

SGs architecture has been seen as an ensemble of devices with problems to be fully integrated and be adaptive to the dynamic of electric power distribution networks due to the kind of protocols used in sensors devices that could lead to communications problems. Therefore, Caballero et al. [67] proposes to use the Web of Energy (WoE) through Actor Model paradigm for the design of infrastructure (concurrent and distributed systems) to support smart functions like resilience, renewing the traditional energy generation, and regeneration of the system, always keeping in mind devices and software. The application of that model gives as a result a middleware architecture for a universal sensor network allowing to have distributed intelligence, real-time data mining, improvement in automation systems and network control functions, simplest configuration and reduction of recovery, and self-healing times in systems.

The adequate monitoring of SGs implies an effective monitoring system by using engineering technology (inspection robots) in combination with wireless sensor networks to get an efficient dynamic monitoring network for transmission lines. Therefore, Fan et al. [68] design a dynamic network coverage and a network nodes deployment method with a coverage graph through a mathematical modeling of dynamic barrier coverage (DBC), the constrains definition can set the dynamic sensing coverage characteristics and needs of linear areas. Furthermore, they present a network coverage algorithm (DCGA) that uses a distributed learning method to analyses each node (calculating and updating utilities) and execute actions according with the utilities of each node and the needs of a determined scenario. This proposal was probed via multiple simulation scenarios (three different scenarios with eight related network performance parameters), showing that DCGA adjusts network execution of dynamic sensor network for transmission line. It also sets the node deployment density to regulate network connectivity and it regulates delay in network by using previously connectivity factor according with requirements of application scenario. Therefore, that proposal balances network monitoring performance and financial cost.

Li et al. [69] expose the importance of multicast routing in multiple applications and developments in SGs as well as in the protection and monitoring of Wide Area Networks (WAN), in order to achieve delays of low duration or to approach to real time. To enable that various applications, protocols and intelligent network tools coexist efficiently, the network must make more efficient communications in the same way with low propagation delay and with approaching to real time. In practice, it is difficult for multicast links not to get congested because they significantly increase network latency. In the literature, everything related to propagation subtractions has been widely studied in a general way, but it is difficult to particularize its use within an Electrical SG. This problem is solved by using the BCBT scheme, defined as a bandwidth ratio tree approach together with a multicast scheme known as SPT (Shortest Path Tree).

SGs reliability depends on the continuity of providing electric energy power. Thus, any rupture of fault situation on the grid must be detected on time. For this purpose, it uses protection devices. Such gears have to offer the ability to detect, measure and adapt protection settings in automatic way, according to each outage situation and the way of communicating this data to each other and with the networks breakers to share optimal control actions and configuration. Analog electro-mechanics relays and modern intelligent electronic devices are used to expose any failure on the grid, but their settings do not adjust while real-time operation takes place. Therefore, Alonso et al. [70] propose the design of new relay based on the IEC61850 smart sensor for coordinating optimal operation time between sensors. This protection scheme was tested in diverse short circuit scenarios and energy penetration levels in a standard SG, reducing the activation time of the network breakers and making the grid more reliable, respect to the analogue and intelligent electronic.

In this paper, Dowlatshahi et al. [71] propose the use of a GMA (Grouping Memetic Algorithm) to address the Set K-COVER problem (maximum number of subsets that covers the network´s needs scheduling sensor´s activity) through running out experiments with diverse targets and number of sensors that show a lifetime increase not only in wireless sensor networks also in IoT and consequently in SGs.

Quality of service on SGs in [72] is guaranteed by Faheem et al. using CARP (channel assignment and detection channel and forwarding algorithms) to get information from IMWSNs (Industrial Multichannel Wireless Sensors Networks) about changes in distribution process and power generation that must be gathered and transmitted to the control center in reliably and efficient way despite environment or conditions of SGs. Experimental comparison of CARP with two already existing schemes (G-RPL and EQSHC) used in SG were carried out with dataset analysis getting from IMWSNs and showing this data are valuable for algorithms validation even if intentions is the design or development of new ones.

Huang et al. [73] present a novel algorithm for a location of Wireless Sensor Networks in a three-dimensional plane. This model is based on the basic theory of the IoT with the help of the DV-Hop model, based on the location of three dimensions (3D) by vector distance hopping which is basically an improvement in the accuracy of 3D location of DV-Hop, better known as 3DDV-Hop. When an A * algorithm is combined with 3DDV-Hop in an intelligent sensor network, an MA * -3DDV-Hop is established. Thus, the algorithm proposed by the authors improves the efficient management of the location of the nodes in the z-plane, correcting the average distance from each jump. Another algorithm that is integrated is a genetic algorithm that classifies with a multiobjective optimization or NSGA-II, who oversees estimation improvement of spatially neighboring coordinates.

Based on the loT and using the Fast Fourier Transform, Kumar et al. [74] developed a smart energy meter to monitor and control the effective energy consumption along of a SG to avoid distortion of current and voltage waveforms that are generated from the ATMega328-P microcontroller. In different nonlinear load conditions, this power quality meter allows to reduce deviation of the smart energy meter when the injected harmonics increased because of the ESP8266 module is used for monitoring by connecting the IoT server with the harmonics analyzer system. As future works, it could be found low-cost controllers suitable for the smart meter, future energy utilization patterns, demand projection, and energy conservation solutions from the stored data in the cloud.

Sanjay [75] proposed the multicast probabilistic model using LA-ANA with quantified metrics in MAC layer to avoid congestion and dark spots in Wireless Sensor Networks for getting to back-haul network with reachability of specific nodes and routers in a SG. This proposal helps in routing packets within the neighborhood along with updating their dependency address, avoiding performance anomaly and increased throughput with higher fault tolerance levels.

Integration requirements (standardization, response time, and security) of electric power networks equipment from power substations to SGs could be problematic as standard communication protocols between grid’s devices are different. Any upgrade is expensive and requires a comprehensive analysis of the needs for successful integration. De Araújo et al. [76] proposes the use of wireless sensor network (WSN) sink node of a Zig Bee-based WSN as a communication protocol link. Electrical devices have a sensor node in which is running a middleware enabling the conversion of data between power substation control center (PSCC) and electrical equipment in the SG. Experiments with power meters (SG elements) wired to sensor nodes allow to get customer energy consumption date, showing that interoperability is done quickly and safely, and any new sensor done can be automatically configured.

Due to environmental monitoring, health care, mart grid, and surveillance, among other applications, used wireless sensor networks, a significant number of investigations have proposed security protocols, exhibiting serious security flaws. For example, Ryu [77] found two serious security weaknesses in authentication schemes: it could lead to user impersonation attacks and the anonymity of the user was not preserved in their scheme. Therefore, a new scheme was developed by Ryu [77] in order to complement their susceptibilities, as well as to improve and to speed up the vulnerability of authentication schemes. In addition, security analysis was performed by Proverif, and informal analysis was carried out for various attacks, finding a significantly better improvement than the existing user authentication schemes.

According to Kaplan et al. [78], communication in SGs is bidirectional between consumers and providers, allowing energy flows to optimize not only network efficiency but also reliability and sustainability. In the future, energy generation and distribution systems are crucial to estimate how much energy is needed to sustain the electricity grid and to calculate user consumption. Therefore, there has been an accelerated evolution of SGs due to the adaptation of energy markets, increasing the complexity of every element of an electrical network, and therefore, including renewable energies in electric vehicle systems and in control of inverter devices. The main challenge is for all actors to maximize their profits or savings, e.g., consumers will see their rates reduced due to high competition while distribution companies will increase their profits due to a greater number of users without increasing infrastructure. However, there are failures generating extraordinary costs due to maintenance. These failures can be avoided with more effective monitoring systems based on artificial intelligence tools with a predictive approach for allowing the supply to not stop and an adaptation to the various conditions of a network. Thus, the authors propose (i) algorithms for forecasting the real load due to consumers based on the analysis of data collected with deep learning and (ii) algorithm for predicting anomalies by distributors of electrical energy to make a natural transmission to more modern systems that evolve according to the real needs of the network.

Saadat et al. [79] emphasize the importance to protect the network control system to avoid or minimize attacks in their integrity, availability, confidentiality that could lead to economy, human, safety and health losses. Therefore, the author proposes a methodological approach for mitigating controls by using an analysis of the past and the future on cyber-attacks to SGs.

### 3.6. Not Defined

Liu et al. [80] propose to use a Markov decision process for allowing consumers in SG decide the right time to schedule buy or sell electricity according with their system´s current state. It enables to maximize net profits and decrease sunk cost when the sold electricity as well reduces the purchase cost and the electricity consumption goes up. This point of view on consumer´s behavior consider that the amount of electricity generated is according to consumer´s infrastructure, weather, time of day and initial investment so that it is possible to predict future levels of electricity generation or demand using weather forecast and historical data. Therefore, benefits are maximized considering at the same of time-pricing variation, actual storage status, appliance usage and peak periods for each specific decision interval. According to the current state of consumer´s system, every resolution leads to an action and in turn to another consumer´s behavior, making possible execution of extensive simulations and comparing the results with competition scheme baseline.

Fenza et al. [81] present an approach based on machine learning to distinguish drifts from real anomalies on SGs. Therefore, a network is trained with different consumer profile to get a model for predicting levels of electricity consumed at certain time-lap and compare it with the real usage. This provides the capacity of calculating a prediction error and a standard deviation range to point out a real anomaly, getting earliest alerts to minimize energy and non-technical wastes via the analysis of real-time data of power consumption obtained from smart meters that continuous monitoring consumer activities and demands of energy. However, that proposal has the inability of recognizing anomalies due to the lack of previous consumer observations in the first week of system´s functioning and also it has a delay between the anomaly happening and its detection, affecting the system’s overall performance in consumer behavior detection and analysis.

Attacks on SGs of information are a threat to their security and stability. Oozeer and Haykin [82] establish these attacks have cyber and physical elements. The system state estimator and bad data detection on SG are vulnerable to the last one, as if the attacker has knowledge of the method and threshold used to detect Bad Data Injection (BDI), it easily can insert false data leading to wrong system state estimations, as well as an improper operation performing and incorrect control decisions. That work focuses on a new model system for control and attack detection based on Cognitive Dynamic System (CDS) model, using a threshold that evolves during every perception-action cycle (PAC) and uses Cognitive Risk Control (CRC) as a special function to detect, control and mitigate abnormal uncertainty or FDI. Also, dual cognitive controller (Task-Switch Control, TSC) is proposed for working together to handle sections of the grid that are under attacks and those ones that have presence of normal uncertainty. When this deviation no longer represents a risk, this cognitive controller can switch off the cognitive risk control. This TSC can change to an executive or cognitive controller with novel set of actions that affects the system configuration and changes as the situation attacks does.

Molina [83] establishes the importance of microgrids to have full potential from all elements of SG, particularly distributed renewable generation energy resources that depends on non-stable and interrupted sources. To put all this together and secure stability and electricity supply on SG in the best mode, Molina focus on integration of energy storage technologies and electronic power conditioning systems as interface to electrical grid to regulate raw energy form energy storage systems. The principal technologies presented consist of mechanical, electrical, electro-chemical, chemical, and thermal each with its corresponding power conditioning system and contrast them in terms of power rating, storage duration, cost, applications, impact on environment, and lifetime. In general, none of these technologies fulfill the requirements of SG on its own making necessary a beforehand study an analysis for each application in order to determine the suitable mix. Therefore, they achieve cost reductions, less investment on upgrades, integration of renewable energy sources to the grid, reduced emissions, secure energy, less need of foreign energy, decreasing the number of outages and its cost and increases the opportunity of sell excess of energy from grid.

Pop et al. [84] propose to match demand and production of energy in smart energy grids by means of decentralized blockchain mechanisms to provide control distributed management and validation of demand response occurring in low and medium voltage. For secure data of metering devices, a blockchain distributed ledger is used. Moreover, for the fulfillment of each distributed energy prosumer in demands response programs, a self-enforcing smart contract allows establish the energy flexibility forecast, do math referring to the loss of balance in energy on the grid in order to set balancing rules, secure the desire demand energy for each customer profile, as well as calculate rewards and penalties in energy consumption. Approach authentication was made via the implementation of a prototype, using energy data sets of several buildings from blockchain literature. Thus, the SG is capable of making adjustments in near-real-time on energy demands and cost reduction on energy transactions as an established decentralized energy trading mechanism without any third-party intermediary for making calculations for energy consumption prices.

An agent-based approach promoted by Singh et al. [85] improves the network performance and communication reliability on the microgrid network, as enhance intelligence was validated via analysis of performance metrics (variations, queuing, delay, and throughput), obtained for different microgrid scales (small, medium and large). This approach was compared versus multi-agent-based Bellman routing and Bellman-Ford algorithm in fault and traffic model. Routing focus on calculations of the shortest path to reach a specific destination to transmit and communicate generation to distribution data and improve network quality in terms of performance metrics and communication reliability.

The increment of energy demand around the world has made necessary control electric energy utilization. Hence, Hafeez et al. [86] develop energy management strategies that systematically manage the power usage in residential building with IoT. Each way to manage electricity arises from a wind-driven bacterial foraging algorithm (WBFA) for (i) reducing cost of electricity, (ii) securing service in demands highest points to alleviate peak-to-average radio, (iii) making IoT technology sustainable, and (iv) decreasing the customers waiting time. The energy management framework was compared with benchmarking strategies carrying out simulations, showing outperform in terms of operational metrics since WBFA based on energy management controller habilitate with IoT technology takes as inputs: (a) the power rating, (b) length of time operation, (c) price based on demand response programs, and (d) accessible power grid energy, in order to schedule electric power use on smart devices of residential building, considering constraints and function.

Poor communication conditions (power lines, radio, or mobile), large packets of data, and low network availability are the main goals addressed by Negirla et al. [87]. From PRIME protocol power line communication, authors propose to attack the availability of smart meter to a power supplier and in consequence give meters capacity to safety gather and load behavior consumption, as well as full remote firmware update. For data interchange, a model slice data correctly used at application stage—considering power grid´s noise levels and adjusted transmission rate—makes possible stable and full transfers even from far off devices since experimental trials exhibit successful transmission of huge profiling data and devices firmware upgrading. Now, it is possible to develop an efficient SG via suitable equipment for interchanging and analyzing large amount of data to get consumer profiles and using different kinds of algorithms to establish smart meter parameters to overcome consumers necessities.

Tradacete et al. [88] focus on developing a system for providing a two-way connection to make possible communication through the several elements of electricity grid, offering an integration option for devices that already connect and new ones. Transformation of a BTSs in a DC microgrids is realized by controlling energy flow from an energy management system. For such transformation it is necessary a hardware/software architecture implementation using low-cost and off-the-shelf hardware applied to any kind of BTS, in order to distribute communication among them in a more cooperative way, and sharing energy for giving surplus energy to the grid.

Integration of electrical grid´s devices represent a challenge in terms of interoperability for allowing IoT technologies performance in the smart grid domain. Cavalieri [89] proposes an inter-working scheme based on the M2M ontology. The authors considered the common ontology between IEC61850 and M2M for giving place to data interchange among devices thanks to the standard one M2M IPE proposed, as well as a definition of common ontology that enriches standard definition of M2M documents. These two most used communication systems (IEC61850 and M2M) work together through the proposed architecture between IoT and SG. Moreover, a semantic interoperability is proposed to give heterogeneity and common meaning to the data exchanged in SGs.

High-speed bandwidth—as one of the main requirements of SGs—can be fulfilled using Broadband Power Lines (BPLC). Slacik et al. [90] use Power Line Communication (PLC) Technology as a simulation option in BPLC area to achieve complex simulations tools whose focal point is end devices communication. A review about Power Line communications (PLC) shows that NS-3 covers satisfactorily enough simulations. This kind of technology needs to consider several kinds of variables for true simulations for getting behavior on specific circumstances and locations. Simulations with the NS-3 standard in electricity distribution networks was carried out via low voltage (LV) or medium voltage (MV) power line for knowing the capacity and communication bandwidth when considering modulation, power spectral density, frequency band, and kind of cable. The simulation of proposed topology design shows transmission rate at the app layer using UDP/IP protocol, and when it was compared with measurement of the real SG topology of substations resemble was detected. Differences between comparison are due to environment, humidity, ages cables and coupling, meaning that in simulations physical variables cannot be setting from modems.

To achieve the best user scheduling, beamforming and coordination of energy in smart-grid powered cellular networks (SGPCNs) or long-term grid energy expenditure, it is necessary heterogeneous energy coordination (energy merchandising in terms of SGs and energy exchange on base stations). Therefore, Dong et al. [91] suggest a two-scale algorithm (TSUBE energy trading algorithm) for approaching a given value of scheduling, and the amount of energy exchanged and beamforming as control parameters tend to infinity. As beamforming vectors and scheduled user equipment are coupled, authors perform a Lyapunov optimization method, letting beamforming be updated over every slot according to the channel changes and address the reliability issues from scheduling or unscheduled user equipment. Interchange between grids energy expenditure and delay of user’s equipment were theoretically achieved when scheduled user equipment indicators exchange of natural renewable energy variables and beamforming vectors are together set out, meaning a possible reduction of the grid energy expenditure and a more suitable interchange among grid energy expenditure and users’ energy data rates. To demonstrate the performance of proposal, a comparison between WOLPE algorithm and ZFBF or zero forcing beamforming was made, showing outperforming of last one over the first one.

Atat et al. [92], propose a stochastic-geometry based power grid model, used as a tool for strategic planning of monitoring, evolution, and expansion through time of power grids, taking in account physical limitations and boundaries as they reflect spatial joints in a specific region. Their model provides a multi-layer distribution of power metering units via a finite-horizon dynamic program for transmission and distribution power systems. The program takes as a input cluster buses, trees buses and their impact factors for metering distribution targets, and in return it gives the number of metering points. The quantity of uncertainty absorbs in each period important information for regions without or with few smart grid infrastructure. This geometry makes it possible tractable dynamic allocation of metering devices in a sizable city and evolving the power grid in a structural way, considering technical and budgetary limitations.

Information on SGs can be compromised in any of their elements. Therefore, Nguyen et al. [93] suggest to detect malicious messages by logging network traffic in dedicated servers. By using router´s ability of duplication packets received with minimal overhead, they formulate an optimal packet collection problem to reduce the sets of collecting points to gather all critical traffic from grid. Thus, it is necessary to determine the number of router sets required, considering SGs communication network characteristics. Proposal includes three algorithms (one changes with time and two are static): dynamic, highly effective, and scalable. The first one deals with updating this solution in critical traffic dynamics, the second one gives an approximation ratio and the third one gives a constant performance ratio. The three algorithms were evaluated versus optimal Integer Programming formulation, showing that the proposal produced is efficient, ensures competitive performance, and produces excellent solutions.

Sapountzoglou et al. [94] carried out simulations experiments using deep neural networks (DNN) addressing four limitations of previous existing methods (grid topology, capability of detect faults with high impedance, localize faults with limited data and number of sensors and perform more than one identification task) to locate and detect fault in Low Voltage Smart Grid Distribution. Method accuracy was measured in depth of the deep neural networks, fault resistance values, types and location, voltage measurements, load demand, size of dataset, and quantity of measurements proving robustness, and it is appropriate for small datasets.

Grid stability and its maximum power can be determined through Thevenin equivalent circuit even if the grid model is unknown. England and Alouani [95] use individual load measures from smart meters to get better Thevenin parameters and real-time stability index that help demand response methods to avoid blackout or power delivery disruption in the grid predicting voltage instability in advance to react in time. Simulations using IEEE30 bus power system were carry out focus on residential areas but also can be applied on industrial o commercial.

The dynamic need of electric power supply for consumers makes necessary an efficient manage of production and distribution to be aimed via a stable SG network with intelligent systems as part of it. For this, there are many Deep Learning approaches (e.g., gated recurrent units, long short-term memory, and recurrent neural networks) that allow determining the reliability of the grid. In Alazab et al. [96] submitted the use of MLSTM (Multidirectional Long Short-Term Memory) to forecast SG stability and compare it the experimental outcomes with another deep learning techniques. Findings of experiments shows that MLSTM model proves out-performance in terms of precision, receiver operating characteristics curve metrics, loss and fidelity compared to gated recurrent units, long short-term memory and recurrent neural networks.

Qaisar et al. [97] proposed an event-driven adaptive-rate sampling approach for the data acquisition and features extraction. The authors’ goal is to eliminate the large amount of redundant data during the acquisition, transmission, and processing stages in SGs by employing the event-driven sampling because it provides a real-time data compression. As a result, the system attains an average appliances consumption pattern recognition accuracy of 96%, which is logged on the cloud via the 5G network. This proposal can be a solution in contemporary automatic dynamic load management and enumerated billing systems. The event-driven adaptive-rate sampling approach should consider other classes of appliances and other robust classifiers like Random Forest, Support Vector Machine, and Rotation Forest.

Another work that faces the inclusion of full-duplex or bidirectional interactions is the one made by Mollah et al. [19], whose proposed interactions optimize and greatly help the operation and especially the maintenance from the point of view of how the network is managed in terms of to their assets. These maintenance and management actions use tools to diagnose, which are costly both economically and computationally speaking, as intelligent monitoring is a very important task but can rarely be performed in real time. Therefore, the authors state that information must be acquired starting from the intrinsic components of the electrical network itself. Due to partial discharges can be considered one of the most harmful elements that the network can handle, the authors propose a way to recognize these discharges using artificial intelligence tools. Thus, the evaluation carried out by the smart sensors along the SG will help to estimate the partial discharges and the continuous verification of the components found along the electrical grid. The sensors can perform this evaluation both automatically and dynamically trying to keep the most optimal conditions of the network over time, increasing reliability and making the initial investment return optimally.

To detect non-technical losses (NTLs) from consumer behavior in a certain time, Li and Wang [98] propose the conversion of smarts meters data to super images with slight data error or missing. Applying deep learning model (based on neural network architecture in computer vision) in a semi-supervised way to the images, it is provided joint features or characteristics for detecting and classifying a wider range of anomalies ensuring real abnormal electricity consumption detection on SGs. This proposal makes use for NTLs detection not only for energy consumption (EC) data but also to get together several kinds of data from SGs such as electrical magnitudes, technological characteristics, quality of measurements, and GIS data.

Chen et al. [99] apply IoT to power systems to make them sustainable and efficient in current cloud computing paradigm, reducing their cost in data transmission. The authors developed a hardware and software architecture of five layers (device, network, data, application, and cloud computing), making it possible power distribution surveillance, access of distributed power and renewable energy and reliable supplies of multiple forms of energy sources, and also analysis of features and consumption of electricity as well as identification of strange behavior via advanced metering system. Making numerical simulations on data prediction, privacy protection and transmission consumption showed that long- and short-term memory algorithm (LSTM) can predict the power price at edge computing devices, allowing to consumer adjust consumption behavior. According to the authors, Laplace noise is a better choice for privacy protection and the requirements for transmission bandwidth and delays are reduced with edge computing architecture.

Traditional SG data management systems are unable to scale, storage and processing enormous amount of data to be aggregated and analyzed. Munshi and Mohamed [100] proposed a SG big data ecosystem based on the Lambda architecture to execute parallel batch and real-time operations on distributed data as well as an ecosystem that uses a Hadoop Big Data Lake to store different kinds of SG data such as smart meter data, image, and video data to allow data mining in digital image and video processing applications. This proposal was tested on a cloud computing platform on real SG data, suggesting that their proposal can perform numerous SG big data analytics.

Bose [101] proposed the usage of modern power electronics and its applications on renewable energy systems and SG due to their impact on power systems, pointing out the relevance of power electronics in solving the climate change problems faced by our society.

Yang et al. [102] proposed a spectrum aggregation-based MAC protocol, called SACRB-MAC (Aggregation Cognitive Receiver-Based MAC), to improve the throughput of the Cognitive Radio Sensor Networks (CRSNs) in SG due to the harsh wireless environment in it. Besides, SACRB-MAC improves the reliability of CRSNs by exploiting the broadcast nature of the wireless medium. Analytical and simulation results demonstrate that SACRB-MAC has a high-capacity and reliable performance, providing an optimistic solution for CRSNs in fulfilling the vision of SGs.

In a system of a SG, the data collected from different SG devices show that they are susceptible to attacks, resulting in disrupted or imbalanced energy between the energy providers and consumers. Alfakeeh et al. [103] proposed a policy-based group authentication algorithm to preserve the security of the demand response in a SG and to provide an access control feature, entering to a limited number of devices connected to SG via the utility server. The first authentication includes a single public key operation and the next ones with or without the same device or other ones without a public key operation, reducing the overheads of communication and computation. This implies a fewer time to successfully establish a secret session key for sharing sensitive information over an unsecured wireless channel. After comparing the proposed algorithm with the state-of-the-art by using formal and informal security analysis, the authors claimed that their algorithm is more resistant to more attacks than the second one.

SGs comprise various embedded intelligent technologies to enhance the safety and the reliability of power grids. SGs exchange information by applying communication protocols like the Open Smart Grid Protocol (OSGP) but failing to integrate with devices compliant with the Constrained Application Protocol (CoAP). Viel et al. [104] proposed a mapping interface integration between OSGP and CoAP, called COIIoT, to enable the communication among IoT devices used in home and industry applications and SG infrastructure. COIIoT also translates the methods utilized in every protocol by means of a set of mapping functions at low-cost and low-impact in communication. Moreover, this proposal reduces the necessary time to develop applications, as the complexity implicated in the communication between the protocols is abstracted.

The use of IoT devices and technologies are capable of supplying to SG the following features: real-time monitoring, advanced pricing mechanisms, dynamic energy management, and self-healing. Nevertheless, changing the traditional grids into SGs makes their components and services prone to cyber-attacks. Diamantoulakis et al. [105] investigated the optimization of attacks and the defending system by the interaction of an attacker and a defender i.e., the number of real devices and honeypots, respectively, focused to maximize their individual payoffs. The authors derived both the Nash Equilibrium (NE) and the Bayesian NE for the first and the second game, respectively, as the corresponding conditions with uncertainty about the payoff of the attacker, as well as offering an alternative structure for situations without equilibrium to optimize the worst-case scenario. Therefore, in an nonexistent equilibrium the attacker can force the defender to receive when they acknowledge the defender’s action for maximizing of its lowest value. The simulation results validated the equilibrium for the attacker, the defender, and for both games. Last, regarding the repeated game, the defender successfully identifies the attacker’s sort, maximizing its payoff along the game.

An online control approach for real-time energy management of distributed energy storage (ES) is developed by Zhong et al. [106] in such a way that each user manages its own virtual ES (VES) without knowing detailed operations of the PESs, via sharing and reallocating their capacities of physical ESs. Applied the Lyapunov optimization framework, this proposal only makes decisions based on the recognition of current system states, without having to predict the future of electricity price, user load, and renewable generation (uncertain system states). According to their algorithm performance, the authors also optimized the offline parameter selection to guarantee the users’ privacy when they send their data to anyone, to manage their VESs locally. As a future work, the authors suggest to designing the pricing rule for the ES sharing service.

Intelligent sensors have been used extensively in SGs, resulting an increasing interest to collect large-scale and fine-grained SG data, in which outliers exist pervasively, originated by system malfunctions, environmental effects, and human involvements. Outlier mining allows to find abnormal patterns from real records that indicate rare and atypical are also recognized as outliers in power generation, transmission, distribution, transformation, and consumption. Sun et al. [107] carried out a comprehensive and systematic re-evaluation of outlier data treatment procedures in the SG environment from the angle of data driven analytics and data mining methods, as well as information security technologies. They provide the usage scenarios of outlier denial and outlier mining in SG environment, concluding that security and reliability of power system operation are the more crucial challenges of outlier data treatment toward smart energy management.

Kazerani and Tehrani [108] propose a new paradigm for smart cities considering old structure of power generation features, fictitious dynamic boundaries implementation based on self-adequacy criteria and control hierarchy of grid of microgrids. Based on the above, the authors expect (a) higher power quality, (b) security, reliability and resiliency in energy system, (c) integration of many energy suppliers in the form of energy hubs, (d) develop and penetration of renewable and sustainable energy sources, and (e) low cost of energy for costumers. According to the authors, their proposal allows self-healing as a new characteristic of SGs to be applied into already existing cities and new ones.

Slootweg et al. [109] pretend to support the conservation of the environment by means of analyzing the negative impact of fossil fuels due to an energy distribution is crucial for our sustainability nowadays. There are many ways to generate renewable energy, such as tidal, solar, and wind. The natural growth of human populations generates a demand that must be met through a generation, supply, and consumption, i.e., sustainable while a bigger number of users are connected to the electrical networks. On certain occasions, two types of certain abnormalities appear in the network: (i) there is surplus energy production and (ii) high consumer demands. These abnormalities are tackled with a balance in the entire electrical distribution network via the implementation of electrical networks that intelligently detect these abnormalities, motivating the need for Smart Networks arises. Therefore, SGs are intended to both control and measure the trend of consumption and their possible consequences before a collapse in the electricity grid occurs.

Beidou et al. [110] analyze six sort of challenges faced by the migration of power system from the conventional grid to the smart grid, and then they propose future research directions to build and implement a reliable, safety, and highly efficient power smart grids as well as possible solutions to tackle today’s energy market challenges.

## 4. Discussion

Regarding Smart Grid Network Topology used or defined by the algorithm, we can see a very homogeneous classification among the algorithms. Thus, as can be seen in Table 1, the schemes are mainly found in Neighborhood Area Networks (NAN) and Field Area Networks (FAN) and Software-Defined Networks (SDN).

Figure 4a shows that most of the articles analyzed are Not Defined (36%) based on what type of network topology they use. Note that 17% can find both in NANs and WSNs, respectively, pointing out a great opportunity to develop smart grids with software-defined topologies.

Table 2 shows the articles that can be classified into Smart Grid Technology. From this table it can be noted that most of the algorithms are categorized into the Internet of Things or Industrial Internet of Things. It is also observed that technologies tend to use artificial intelligence algorithms or the blockchain.

Figure 4b shows that 35% of the algorithms found are in the Internet of Things classification, and specifically 14% are for the Industrial part. Another important point is that one out of every five algorithms make use of some Machine Learning tools and tools of artificial intelligence have gained strength in these proposals.

Table 3 shows that most of the Encryption issues do not clearly define the algorithms they use, although there are some articles that use Multi-dimensional Data aggregation and even Cognitive Risk Control.

Figure 4c shows that most of the algorithms do not include an encryption tool in their methodology, this is the case of 84% of the articles. There are two types of encryption found in the rest of the algorithms described: (i) Multidimensional Data aggregation and (ii) Cognitive Risk Control with 13% and 3%, respectively.

From Table 4, when a classification is made according to the type of current that the smart grid manages, most of them are algorithms, as expected they are alternating current. Although there are some schemes that contemplate the Direct Current for very specific applications in the Industry.

In terms of the Type of current transmitted by the Smart Grid, Figure 5a shows that 86% of Smart Grids networks use or carry Alternating Current (AC). This is expected because almost all final users take 100V/200V AC for domestic or industrial usage. Furthermore, only 14% of the articles founded use Direct Current (DC) as part of their algorithm.

While Table 5 clearly shows us that the schemes presented here mostly contemplate traffic and data transmission over the Smart Grid, regardless of the current for which they are designed.

Approximately three out of every five algorithms found execute a Data Transmission over a Smart Grid. Therefore, Figure 5b shows that 62% use the network to send or receive information, while the remaining 38% do not consider it.

Then, when articles are classified by their Application, a majority focus on Distributed Energy Resources, Anomaly Detection and Privacy Preserving, Table 6. If this table is compared with the others, we can see that it is very homogeneous and therefore the applications and opportunities are very wide.

Figure 5c shows the Applications of Smart Grids. This figure indicates that 60% of the algorithms are related to the detection of abnormal events and preserve privacy, the first with 31% and the second with 29%. There are other algorithms that help remote protection of certain places such as homes or industries, and others that distribute their automation, with 10% and 14%, respectively.

Table 7 shows that Technologies used in Smart Grids are varied, but mostly focused on Quality of Service and WSN. In this classification, few articles talk about more recent connection technologies such as 5G; this can be seen as an opportunity for future research.

According to Figure 6a, the Connectivity used in the Smart Grid, technologies are varied and abundant, since 28% concentrate seven different ways of establishing the connection between members or nodes of the Smart Grid, most of them wireless. Besides, 30% is located in connections of Distributed Sensor Networks and 42% to the Quality of Services of these networks.

The state-of-the-art of the current schemes presented here uses various analytical tools such as the Time Series or the Regression Models, but in most of the works classified in Table 8 do not define what analysis time they did, much less what scheme that is used.

Mathematical analysis tools such as the Time Series or the use of Regression Models should be of utmost importance in terms of Tools used for the analysis of Smart Grids. Therefore, Figure 6b shows that only 13% make some analysis of the information founded and the vast majority with 87% do not use it or do not refer to it.

Finally, Table 9 shows that there is heterogeneity in terms of the algorithms or protocols used, since 22 subclassifications were found, but each of them contains a single algorithm in most cases. Their purpose varies from detecting malicious information or attacks to the grid, connection between smart devices, efficient energy management, secure data transmission, overall performance of electrical and smart grid, better periods for sale and buy energy, architecture design, security, and integration of grids. The works of Alonso et al. [70] and Cavalieri [89] using the IEC61850 Protocol can be highlighted, showing us an applicable field of research with a view to homogenizing the algorithms and protocols used by Smart Grids and achieve an integration of all elements that conform them.

Furthermore, Figure 6c shows the Protocols Applied in Smart Grid Algorithms, pointing out that these technologies are varied and abundant, since 46% concentrate 14 different protocols applied among members or nodes of the Smart Grid. In addition, 34% is located in wireless connections of Sensor Networks and 20% Algorithmic Approaches inside these networks.

## 5. Smart Grid Trends

Due to the optimization of networks and the waste of energy, in this century is crucial to supply energy to cities, reducing their ecological footprint. Thus, Smart Grids (SGs) tend to promote the electrification of demand, knowing that electricity is the most efficient energy vector and respectful of the environment due to the absence of emissions at the destination, as these intelligent algorithms improve many of the existing systems and promote the development and evolution of existing services, as these proposals take advantage of renewable energies.

There are many key elements in the proposal, promotion, and implementation of SGs, highlighting public and private institutions, electric companies, and especially the users of the network. Among these factors we must first highlight the continuity in the introduction of new technologies in the electrical networks, the increase in energy efficiency or the regulatory requirement as support for the change of the energy model. Second, changes in the consumption habits of network users, a greater integration of renewable distributed generation and storage, are also important. All of the above, without forgetting the social aspect, incorporating the citizen at the center of the decision model on which model to cover their energy demand and that of their own city should be built. On many occasions, implementation of SG paradigms exposed in this work depend directly on teamwork between all Government Orders, Electric Companies and representatives of civil society [111].

Thus, from the point of view of energy efficiency, Smart cities and SGs will provide important services to society, mainly focused on saving energy, which will greatly reduce $CO_{2}$ emissions to the environment. Therefore, the services are mainly summarized in two main concepts: (i) Availability of consumption information and (ii) Energy management capacity. All this in order to improve environmental conditions and above all the sustainability of the cities of the future. With the support of all the infrastructure emanating from the installation and development of SGs it can be said that now all of us can have Smart cities, where customers will have advanced information on the electricity consumption of inhabited homes. All this due to the evolution over time, as well as a comparison of your consumption with that of similar users and personalized advice to reduce electrical consumption in an intelligent way. These services give a way to the possibility of acting, planning and managing consumption, interacting on manageable loads, especially in an autonomous and intelligent way. It can also be said that the Energy Service providers will offer their clients more services and systems for the active management of their demand [112].

Moreover, micro and small businesses as well as housing complexes concentrate many buildings and public spaces to be important beneficiaries of these information services on consumption and energy management [113]. Furthermore, they will improve their consumption and energy efficiency, translating this into corresponding savings for society and an improvement in the quality of life in these Smart cities. In addition, the technologies and systems that are installed in these projects allow the integration into the distribution network of many more generation systems of renewable origin, which is a basis for the distribution network of the future. All this also produces environmental benefits. Likewise, the concept of micro-generation and micro-storage, as well as the control algorithms that make it possible, will allow to consider the possibility of self-supply or exporting part of the energy generated in the company to the distribution network towards a home microgrid. Giving intelligence to the electricity grid and the possibility of management will allow an intelligent integration of self-consumption.

Additionally, programming used to meet the objective is multi-level influencing, taking into account the personal data and consumption profiles of users to prevent fake news from reaching them by means of the using of social media services. Various models are also considered to respond to the different attack tactics and how an operator or distributor of the electricity grid responds.

In general, the development of the SGs algorithms defined in this work have focused on the adoption of various technologies that help the electricity sector to mitigate network capacity problems, trying to reduce energy losses and thereby achieve greater efficiency in the provision of the service [114].

All in all, the following benefits can be highlighted:Better network monitoring.Distribution of service interruptions and the total number of affected customers.Timely and reliable fault management.Real-time network reconfigurationOffer of new services.Control of electricity flow in the network.

According to the above, it can be summarized from all schemes described in this work tht can change the traditional model of energy supply generated by the implementation of Smart Grid technology not only improves the management of the distribution system, but also creates value for the final user or costumer. These benefits are obtained from the adoption of the following four trends related to smart electrical grids that are beginning to be used in current distribution systems:Smart metering: Advanced metering infrastructure is helping these algorithms quickly locate service outages and have greater control of energy demand. Moreover, users receive better rates that help them modify their consumption patterns and reduce the value to pay on the bill [115].Green energies: Most of the algorithms summarized here are designed based on programs that, since they integrate local power generation sources into the main grid, relying on service users who have the capacity to inject renewable energy into the grid. Through different compensation schemes, both SGs and customers benefit from this synergy [116].Home energy management: The expansion of the internet of things schemes has reached the electrical appliances in homes, making it possible to administer these from applications that make relevant information about their consumption, rates and connected devices available to users [117].Demand response programs: The participation of end users in the energy supply system has become a fundamental axis to ensure the availability and quality of service during critical moments. Taking advantage of the massification of smart appliances, SGs have greater control of demand in exchange for economic incentives for customers [118].

It is important to highlight that SGs are attentive to the evolution of Smart cities as well as their benefits generated for the industry. Likewise, they must consider the specialized solutions that have advanced measurement functionalities and customer experience allowing them to take advantage of the opportunities that arise during their technological transition towards smart electrical grids.

Smart Grids technologies used had to face many challenges in fields like demand, consumption, and energy creation. It implies a continuous searching of the best options to transmission capacity to add more renewable resources like wind or marine technologies and make small systems (houses and buildings) capable to be efficiently operational with larger system. Thus, a communication infrastructure must keep the whole system sharing information in two ways making it possible an operational grid energy generation, storage, and trade. This challenges on its own make necessary all the actors play an active role in the operation of the system through real time information analysis. Moreover, these actor must be regulated by policies that helps educating consumers, guarantee data cybersecurity, establish prices, conditions to sell and buy, asset utilization, prevent energy theft, less intermittency in service or blackout.

Actual papers do not address the creation of models to promote the investment among countries over the worlds to build and optimize infrastructure and continuously develop better grid architectures, advanced components, monitoring and predict models. Despite the fact, some of them talk about a profound analysis of the system to reduce power purchase and production costs. Furthermore, a detailed inspection on topics like electric utility operating expenses, IT and data management systems, energy-saving devices and above all reduced power plant emissions, allowing environmental-climate preservation and energy sustainability. Besides, we find that the ten main problems of current systems can be the following:Cyber-attacks.Low power and lossy networks.Coupling two networks to simulate a real network.System malfunctions and environmental effects when collecting large-scale and fine-grained SG data.Scale, storage and processing enormous amount of data to be aggregated and analyzed.The dynamics of home area network SG communication system.The harsh wireless environment in SG.Serious security flaws in SG.The increment of energy demand in large-modernized cities.Cyber-attacks causing disrupted or imbalanced energy between the energy providers and consumers.

On the other hand, the top ten opportunities that Smart Grids provide can be listed:Efficiency in energy consumption.Green communications in SG.Coupling a cyber physical and a social network.Outlier mining allows to find abnormal patterns from real records in power generation, transmission, distribution, transformation, and consumption.Allowing data mining in digital image and video processing applications.Power electronics as a tool for solving the climate change problems faced by our society.To improve the management of the transmission of data through a smart meter between communicating devices.Exploiting the broadcast nature of the wireless medium in SG.Security protocols to detect security weaknesses in SG.Emergence of microgrids with sensing technologies and Fog-Cloud computing infrastructures to build smart electrical grids.

Therefore, five main advantages of Smart Gris are (i) Models to locate malicious node in SG nodes, provide the usage scenarios of outlier denial and outlier mining in SG environment, (ii) Performing numerous SG big data analytics, power electronics as a relevant item in SG and in renewable energy systems, (iii) To model the effect of traffic intensity and the quantity of active devices in home area networks, (iv) To improve the reliability of Cognitive Radio Sensor Networks, and (v) Improving and speeding up the vulnerability of the security in SG.

Finally, the following five disadvantages can be highlighted: (i) High requirements of communication networks to convey, sense and control data; security for primary users, fulfilling the QoS requirements of communications in SG, (ii) It is not designed the pricing rule for the energy storage sharing service in the SG, (iii) There are not low-cost controllers suitable for the smart meter, future energy utilization patterns, demand projection, and energy conservation solutions from the stored data in the cloud, (iv) It is necessary a more complex and interdependent empirical model of sensor networks to simulate actual network systems, and (v) other classes of appliances and other robust classifiers like Random Forest, Support Vector Machine, and Rotation Forest must be considered when collecting and extracting data.

In the case of sensors, one possibility is to optimize the links between HAN to NANs and NANs to WANs for various SG applications, and investigate the suitable topologies for HANs, NANs, and WANs, according to the size of the system or grid with the corresponding design of routing protocols for secure and QoS-ware data gathering from various SG applications. Besides, it is essential to monitoring components that enable to get data for rapid diagnostics and precise solutions for any event, using the devices and algorithms that autonomously take appropriate corrective actions to eliminate, mitigate, and prevent any power quality disturbances.

## 6. Conclusions

In this paper, a taxonomy of nine classifications about current Smart Grid algorithms was built, showing effectively the energy requirements of both consumers and distribution companies. This helps all the factors involved in the administration and distribution of electrical energy to reduce the gap between demand and supply. It is important that Smart Grid systems and schemes are not new concepts; however, the pace of implementation is of enormous importance today. All algorithms have various advantages and disadvantages, but in all there is a need to contribute and break down barriers for the creation of stable, sustainable, and future-oriented schemes.

Besides, an integration of the advanced communication infrastructure is shown in many cases, and it is considered one of the main problems in the implementation of almost all the algorithms classified. An essential part of these systems is the presence of an entity that consumes the services provided by Smart Grids. They are considered as consumers to be benefited from the great variety of devices to be able to communicate with each other and to be participants in more complex systems needing communication between all the factors influencing these Smart Grids. Infrastructure existing today and allowing communication between the systems interacting in the Smart Grids for now are not capable of fully satisfying the demands for the Commercial Smart Grids to become a reality. Therefore, proposals are required to integrate protocols, applications or diverse technologies making use of more robust analysis tools.

The mainly perspectives to consider in Smart Grid research are conditions or state of electrical energy and control in the electricity production in the system. Nevertheless, the increase of world population motivates the researchers also to consider the peaks in demand and consumption of energy, as well as variations in prices and increase the net profits of suppliers and decrease the lost cost. It is also important to take into account the consumer’s perspective to minimize the total cost or maximize some utility function. In addition, always keeping in mind the operational components and devices for energy measurement, development of a real-time price algorithm where both buyers and sellers benefit. These articles pointed out the needs such as finding a renewable ecologic source of energy and integrate the existing devices and structure of electric power grids to Smart Grid in order to make the best choice to supply energy among countries and stop climatic change. Another topic to attend is a faster policy that regulates the industry of smart grids and make it sustainable in long term.

Thus, these energy consumers will be interested in machine-to-machine communication, i.e., making micro networks of sensors or Internet of Things, needing different communication protocols. Therefore, most of the algorithms expose the concern and the whole challenge the interoperability among the different smart sensors.

For any of these schemes to become a reality in the energy management and distribution market, distribution companies need to obtain detailed and continuous information on users’ consumption and thus intelligently estimate their needs. Thus, user’s information can reveal the most important activities of a household, causing serious security and privacy problems. Therefore, all algorithms should have security algorithms and be resistant to cyber-attacks.

Finally, the growth or massive exposure of these algorithms is seen as one of the biggest challenges in the smart grid environment. This is due to the fact that the number of users increases day by day, requiring more data speed and bandwidth to avoid channel propagation delays. Therefore, the trend of the algorithms presented here is towards the use and adaptation of a better and efficient communication channel.

## Figures and Tables

**Figure 1 sensors-21-06978-f001:**
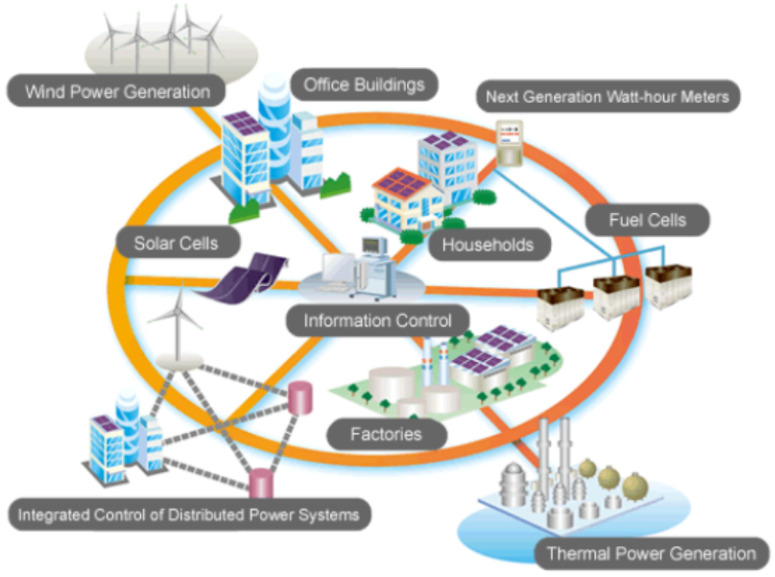
Main factors that influence the composition of a Smart Grid.

**Figure 2 sensors-21-06978-f002:**
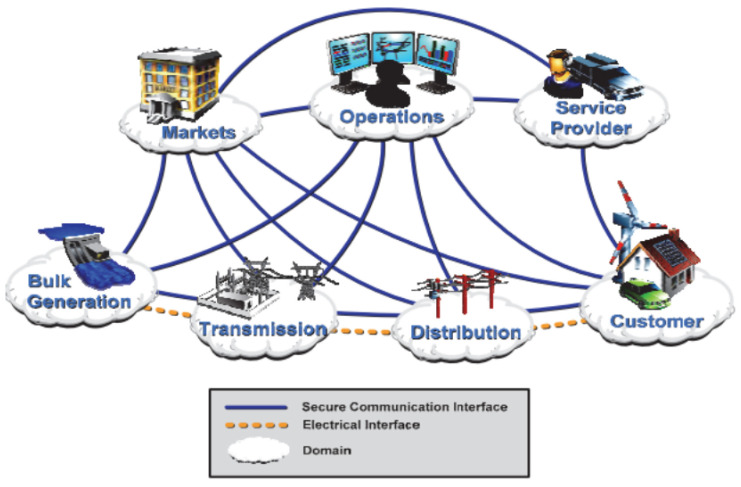
Conceptual scheme of Smart Grids according to NIST.

**Figure 3 sensors-21-06978-f003:**
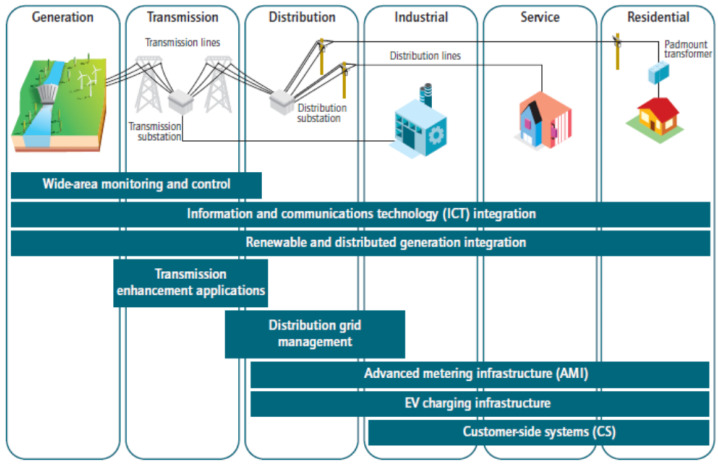
Smart Grids Technologies by subsector.

**Figure 4 sensors-21-06978-f004:**
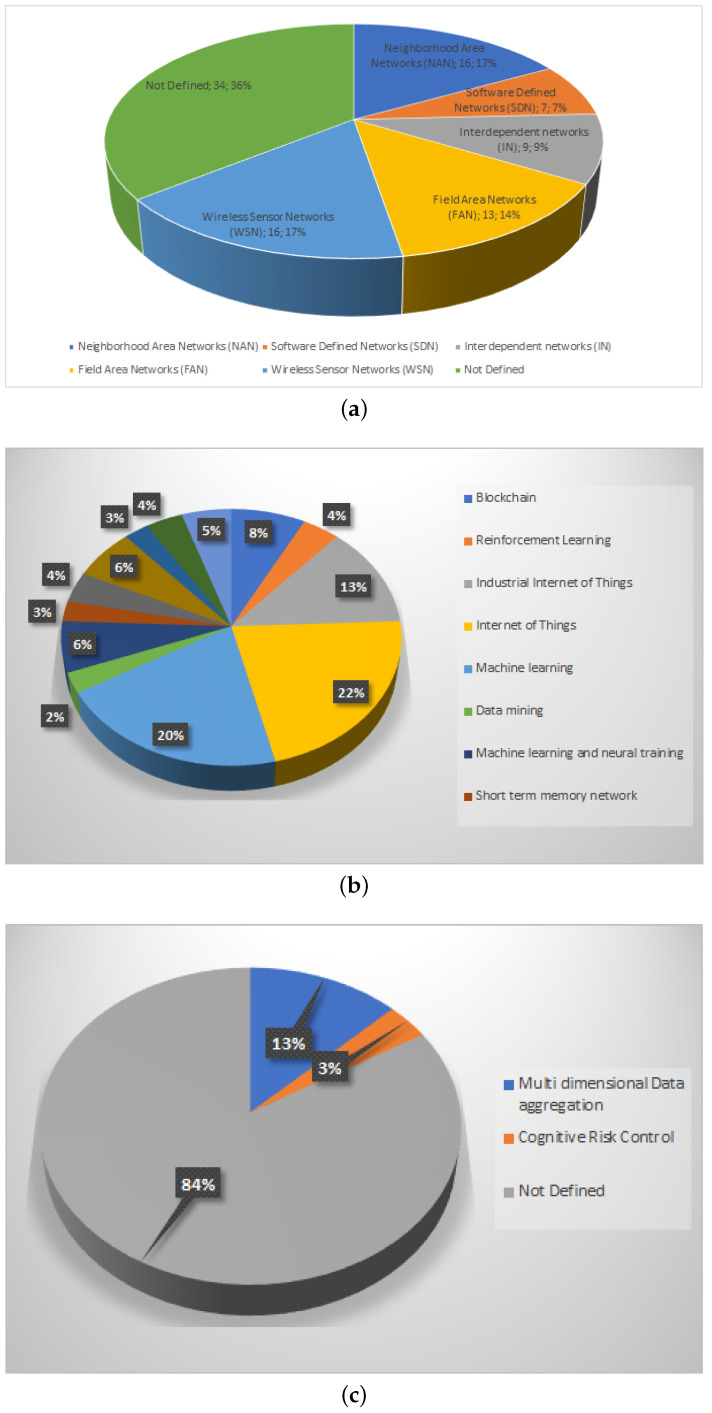
Classification: (**a**) Smart Grid Network Topologies, (**b**) Smart Grid Technologies, and (**c**) Encryption used in Smart Grids.

**Figure 5 sensors-21-06978-f005:**
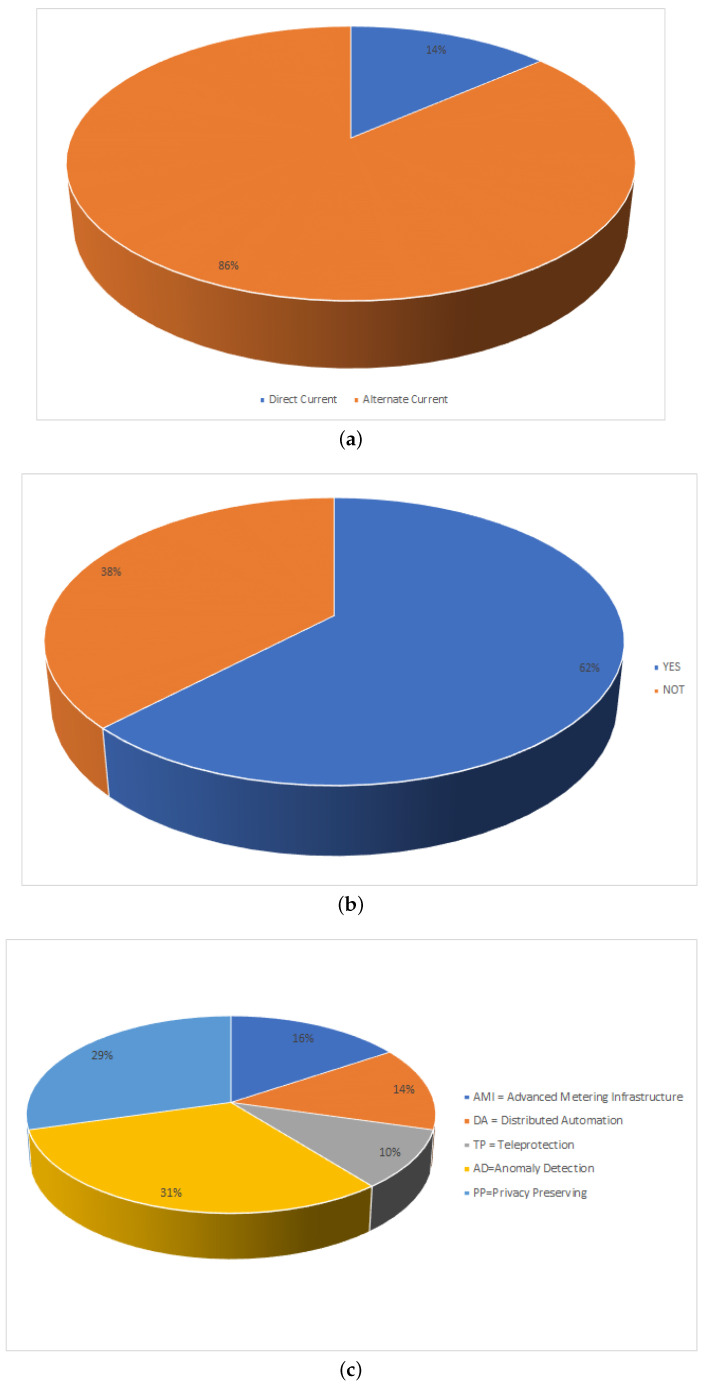
Classification: (**a**) Type of current transmitted by the Smart Grid, (**b**) Data Transmission over a Smart Grid, and (**c**) Applications of Smart Grids.

**Figure 6 sensors-21-06978-f006:**
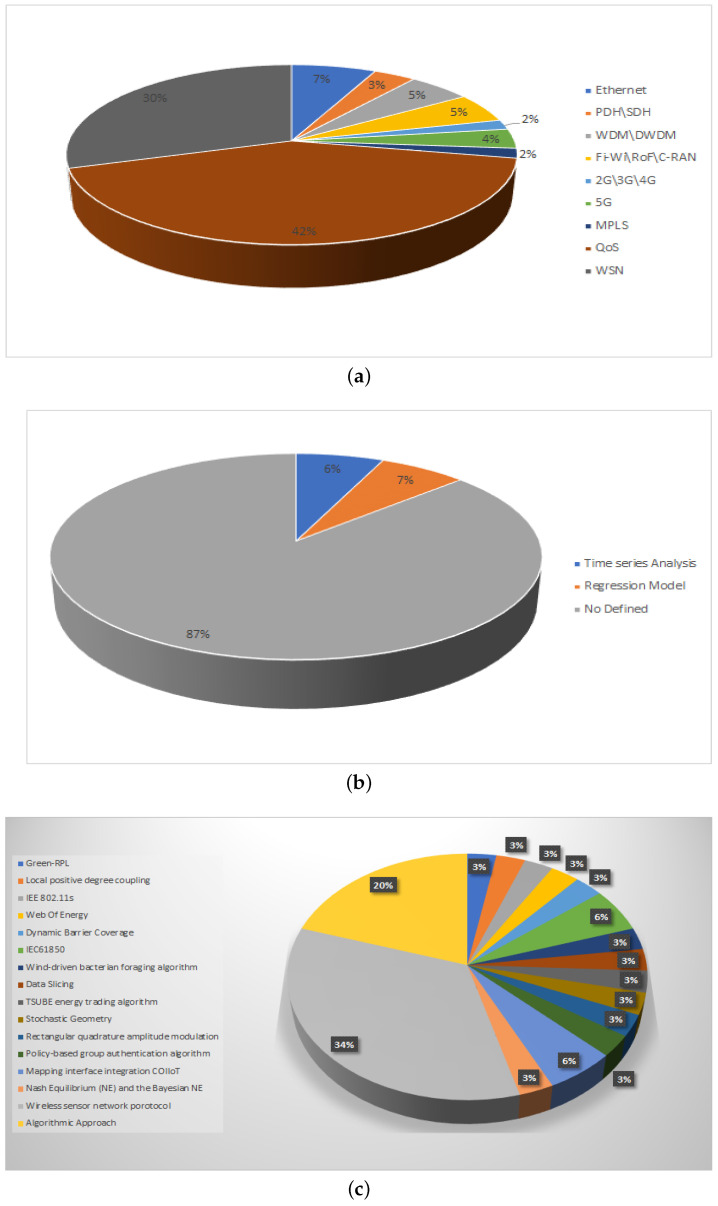
Classification: (**a**) Connectivity used in the Smart Grid, (**b**) Tools used for the analysis of Smart Grids, and (**c**) Protocols Applied in Smart Grid Algorithms.

**Table 1 sensors-21-06978-t001:** Articles that can be classified by means of their Network Topology.

Neighborhood Area Networks (NAN)	[22,23,24,25,26,27,28,29,30,31,32,33,34,35,36,37]
Software Defined Networks (SDN)	[20,38,39,40,41,42,43]
Interdependent networks (IN)	[44,45,46,47,48,50,51,52,53]
Field Area Networks (FAN)	[21,49,54,55,56,57,58,59,60,61,62,63,69]
Wireless Sensor Networks (WSN)	[64,65,66,67,68,69,70,71,72,73,74,75,76,77,78,79]
Not Defined	[19,53,80,81,82,83,84,85,86,87,88,89,90,91,92,93,94,95,96,97,98,99,100,101,102,103,104,105,106,107,108,109,110]

**Table 2 sensors-21-06978-t002:** Articles that can be classified as smart grid technologies.

Internet of tdings	[19,23,24,27,28,30,31,36,37,58,64,66,73,74,79,105,107]
Machine learning	[20,29,32,51,55,56,60,62,69,78,80,81,85,98,102,106]
Data mining	[47,107]
Machine learning and neural training	[21,54,69,81,96]
Short term memory network	[41,61]
Power Line Communication Technology	[53,63,90]
Power electronics	[35,50,99,101,109]
Big data	[100,107]
Fog Cloud computing	[48,52]
Energy Storage and Power Electronics Technologies	[76,83,108,110]

**Table 3 sensors-21-06978-t003:** Articles that can be classified by means of their Encryption Algorithms.

Multi-dimensional Data aggregation	[42,55,56,57,60,62,64]
Cognitive Risk Control	[37,82]
Not Defined	[19,20,22,23,24,26,28,29,31,35,36,38,39,43,45,46,47,48,50,52,53,54,58,61,63,73,76,77,78,79,81,98,99,100,101,102,103,104,105,106,107,108,109,110]

**Table 4 sensors-21-06978-t004:** Articles that can be classified by means of their current types.

Direct Current	[28,48,55,88,99,107,108]
Alternate Current	[19,21,22,23,24,25,26,27,29,30,31,35,36,37,38,40,41,42,43,45,46,47,50,52,53,54,56,57,58,59,60,61,62,63,69,73,76,78,79,98,108,109,110]

**Table 5 sensors-21-06978-t005:** Articles that can be classified by means of their data transmissions over AC/DC.

YES	[20,21,23,24,25,26,27,28,30,32,35,36,37,40,41,43,46,47,52,53,55,56,57,58,59,60,64,69,76,78,79,99,103,104,107,108,110]
NOT	[19,22,29,31,38,39,42,45,48,50,53,54,61,62,63,73,77,98,101,102,105,106,109]

**Table 6 sensors-21-06978-t006:** Articles that can be classified by means of their applications.

AMI = Advanced Metering Infrastructure	[20,21,24,48,53,69,73,107]
DA = Distributed Automation	[25,31,38,53,78,107,108]
TP = Teleprotection	[36,37,45,46,50]
AD = Anomaly Detection	[19,23,29,32,35,43,44,47,52,63,76,81,98,105,107,109]
PP = Privacy Preserving	[30,32,39,49,53,57,61,64,77,79,99,103,104,110]

**Table 7 sensors-21-06978-t007:** Articles that can be classified by means of their Technologies.

Etdernet	[22,30,54,107]
PDH\SDH	[40,41]
WDM\DWDM	[20,27,69]
Fi-Wi\RoF\C-RAN	[23,24,58]
2G\3G\4G	[59]
5G	[21]
MPLS	[56]
QoS	[19,25,28,29,31,37,39,43,45,46,47,50,52,53,55,60,62,78,81,98,99,109,110]
WSN	[26,35,36,38,42,48,57,61,63,64,65,73,76,77,79,108]

**Table 8 sensors-21-06978-t008:** Articles that can be classified by means of their Analytical Tools.

Time series Analysis	[21,23,27,29]
Regression Model	[31,41,54,56]
No Defined	[19,20,22,24,25,26,28,30,35,36,37,38,39,40,42,43,45,46,47,48,50,52,53,55,57,58,59,60,61,62,63,69,73,76,77,78,79,98,99,100,101,102,103,104,105,106,107,108,109,110]

**Table 9 sensors-21-06978-t009:** Articles that can be classified by means of their Algorithms or Protocols.

Green-RPL	[53]
Local positive degree coupling	[44]
IEE 802.11s	[33]
Web Of Energy	[67]
Dynamic Barrier Coverage	[68]
IEC61850	[70,89]
Wind-driven bacterian foraging algorithm	[86]
Data Slicing	[87]
TSUBE energy trading algorithm	[91]
Stochastic Geometry	[92]
Rectangular quadrature amplitude modulation	[37]
Policy-based group authentication algorithm	[103]
Mapping interface integration COIIoT	[99,104]
Nash Equilibrium (NE) and the Bayesian NE	[105]
Wireless sensor network protocol	[35,50,53,63,73,76,77,78,98,108,110]
Algorithmic Approach	[36,37,48,52,93,107,109]

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
