# Peer review of "A Comprehensive Review on Smart Grids: Challenges and Opportunities"

_sensors, 2021, doi:10.3390/s21216978_

Round 1

Reviewer 1 Report

The article of "A Comprehensive Review on Smart Grids: Challenges and Opportunities" describes the research about smart grids. However, some mirror changes are suggest.

1 Some literature seems limited and outdated. For instance, the authors should mention the work in 2021.

2. All the reference in each table need to give to some explanations.

Reviewer 2 Report

Authors have summarized recent research articles in Smart grids with focus on classifying various algorithms/technologies involved. The classification itself is quite perplexing. Here are a few suggestions to improve the review paper: 

  1. Authors claim that the main contribution of this paper is by contemplating a taxonomy of nine classifications unlike the previous review articles. However, the basis for this classification is unclear. For instance, in abstract, authors mention that the paper proposes a taxonomy of many technologies whereas on line 131-136, they state a taxonomy of different algorithms. The authors should clarify this. 
  2.  The first classification is based on Smart grid technologies with sub-classification such as machine learning, blockchain, machine learning and neural network etc. Later, they have another classification based on algorithms such as markov process, fuzzy logic, genetic algorithm and others. It is strange since these algorithms fall under machine learning category. 
  3. Under applications category, the authors include substation automation and Distributed energy resources (DERs) as sub-classifications. It is hard to understand how these are applications of Smart Grid. 
  4. The authors can find better ways to graphically represent the classification rather than listing them out. 
  5. Section 3 is extremely hard to comprehend and is unclear. There are no sub-sections in the 17-page long section. It would be better if the section has sub-sections based on categories previously mentioned. 
  6. The paper requires extensive English and grammar editing. For instance, abstract needs to be rephrased. Other examples are line 51-52, 60-6184-85,130-136, 244-246,315-317378-380,392-393, 1097-1099,1178-1180,1213,2127,1273,1282-1283. 
  7. line 495, a reference is missing (Wu et.al)

Reviewer 3 Report

The article is not well organized, the authors talk about several technologies used in smartgrids . Each technology requires a precise and separate study. No smartgrid architecture is presented. This paper looks like a smart grid keyword summary or report without any Challenges and Opportunities.

In the conclusion part, the authors specified that: "the challenges and problems of 77 current Smart Grid algorithms are discussed" while we find no trace of the algorithms or a table that summarizes the advantages and disadvantages of each technology. 

Round 2

Reviewer 2 Report

Authors have thoughtfully responded to every comment previously mentioned. 

Reviewer 3 Report

Figure 6 (c) is not readable, it needs to be changed.

The quality of the figures is not good (example: Figures 1 and 4).

A review needs a discussion part. The discussion part is missing in this article.

The authors can use the references below to improve their article.

1)     Grid of Hybrid AC/DC Microgrids: A New Paradigm for Smart City of Tomorrow, 2020 IEEE 15th International Conference of System of Systems Engineering (SoSE).

2)     Sensing and control challenges for Smart Grids, 2011 International Conference on Networking, Sensing and Control.

3)     Fault Diagnosis of Smart Grids Based on Deep Learning Approach, 2021 World Automation     

Congress (WAC).

4)     Smart grid: Challenges, research directions and possible solutionsThe 2nd International Symposium on Power Electronics for Distributed Generation Systems

5)     Smart Grid and Cybersecurity Challenges2020 5th IEEE Workshop on the Electronic Grid (eGRID)
